# Developmental, cellular, and behavioral phenotypes in a mouse model of congenital hypoplasia of the dentate gyrus

Amir Rattner[1], Chantelle E Terrillion[2], Claudia Jou[3], Tina Kleven[4], Shun Felix Hu[3], John Williams[1,5], Zhipeng Hou[6], Manisha Aggarwal[6], Susumu Mori[6], Gloria Shin[7,8], Loyal A Goff[7,8], Menno P Witter[4], Mikhail Pletnikov[2], André A Fenton[3,9,10], Jeremy Nathans[1,5,7,11]*

[1]Department of Molecular Biology and Genetics, Johns Hopkins University School of Medicine, Baltimore, United States; [2]Department of Psychiatry and Behavioral Sciences, Johns Hopkins University School of Medicine, Baltimore, United States; [3]Department of Physiology and Pharmacology, Robert F. Furchgott Center for Behavioral Neuroscience, State University of New York, Downstate Medical Center, Brooklyn, United States; [4]Kavli Institute for Systems Neuroscience and Center for Neural Computation, Norwegian University of Science and Technology, Trondheim, Norway; [5]Howard Hughes Medical Institute, Johns Hopkins University School of Medicine, Baltimore, United States; [6]Department of Radiology and Radiological Science, Johns Hopkins University School of Medicine, Baltimore, United States; [7]Department of Neuroscience, Johns Hopkins University School of Medicine, Baltimore, United States; [8]Department of Genetic Medicine, Johns Hopkins University School of Medicine, Baltimore, United States; [9]Center for Neural Science, New York University, New York, United States; [10]Neuroscience Institute at the New York University Langone Medical Center, New York University, New York, United States; [11]Department of Ophthalmology, Johns Hopkins University School of Medicine, Baltimore, United States

*For correspondence: jnathans@jhmi.edu

Competing interests: The authors declare that no competing interests exist.

**Abstract** In the hippocampus, a widely accepted model posits that the dentate gyrus improves learning and memory by enhancing discrimination between inputs. To test this model, we studied conditional knockout mice in which the vast majority of dentate granule cells (DGCs) fail to develop – including nearly all DGCs in the dorsal hippocampus – secondary to eliminating *Wntless* (*Wls*) in a subset of cortical progenitors with *Gfap-Cre*. Other cells in the *Wls*^fl/-^;*Gfap-Cre* hippocampus were minimally affected, as determined by single nucleus RNA sequencing. CA3 pyramidal cells, the targets of DGC-derived mossy fibers, exhibited normal morphologies with a small reduction in the numbers of synaptic spines. *Wls*^fl/-^;*Gfap-Cre* mice have a modest performance decrement in several complex spatial tasks, including active place avoidance. They were also modestly impaired in one simpler spatial task, finding a visible platform in the Morris water maze. These experiments support a role for DGCs in enhancing spatial learning and memory.

## Introduction

The hippocampus plays a central role in memory formation and retrieval. Current evidence indicates that the activities of ensembles of hippocampal neurons represent distinct memory elements as well

as the relationships between those elements. In mammals, the hippocampus is comprised of four principal components: the dentate gyrus (DG), and the CA1, CA2, and CA3 subfields. The most prominent circuit for hippocampal information flow is a trisynaptic loop in which (i) stellate cells in the entorhinal cortex send axons to synapse onto dentate granule cells (DGCs; the major class of excitatory output neurons in the DG), (ii) DGCs send their axons, the mossy fibers (MFs), to synapse onto CA3 pyramidal cell dendrites, (iii) CA3 pyramidal cells send their axons, the Schaeffer collaterals, to synapse onto CA1 pyramidal cell dendrites, and (iv) CA1 pyramidal cells send axons to synapse onto pyramidal cells in the entorhinal cortex (*Siegelbaum and Kandel, 2013*). Additional projections from the entorhinal cortex synapse directly onto hippocampal pyramidal cells.

The function of the DG has been an object of investigation and speculation for more than 50 years (*Treves et al., 2008*; *Hainmueller and Bartos, 2020*). Among its many proposed functions, a role for the DG in pattern separation is broadly supported by data from both rodents and humans (*Leutgeb et al., 2007*; *McHugh et al., 2007*; *Bakker et al., 2008*; *Nakashiba et al., 2012*; *Neunuebel and Knierim, 2014*; *Baker et al., 2016*; *Berron et al., 2016*). These data are consistent with a model in which the DG improves memory discriminations by enhancing the distinction between similar inputs (*Ruediger et al., 2011*; *Sasaki et al., 2018*; *van Dijk and Fenton, 2018*).

Research on DG function has been hampered by the DG's large size and relative inaccessibility within the brain. These characteristics preclude its surgical ablation without substantial collateral damage. While viral gene transfer and optogenetic methods permit controlled activation or inactivation of subregions of the DG, it is impractical with current viral injection and fiber optic technologies to manipulate the entire DG bilaterally. One partial solution to this challenge is selective pharmacologic ablation of DGCs by local injections of the microtubule depolymerizing drug colchicine (*Goldschmidt and Steward, 1980*; *Walsh et al., 1986*). For reasons that are unclear, DGCs are more sensitive to colchicine than are the other major classes of hippocampal neurons. However, the activation of microglia secondary to DGC death (*Goldschmidt and Steward, 1982*), and the likelihood of sublethal physiologic effects of colchicine on other classes of neurons makes this method less than optimal (reviewed in *Xavier and Costa, 2009*). Similarly, X-irradiation of the neonatal forebrain reduces DGC number and impairs performance in spatial tasks, but the interpretation of these experiments is complicated by radiation effects on other brain regions (reviewed in *Xavier and Costa, 2009*). A more recent approach takes advantage of the relative selectivity of a *Proopiomelanocortin* (*Pomc*)-*Cre* transgenic mouse line for DGCs, thereby providing genetic access to these cells with *Cre-Lox* technology (*McHugh et al., 2007*; *Haws et al., 2012*; *Jones et al., 2016*).

The present work describes a new mouse model that can be used to explore the role of the DG in general – and DGCs in particular – in hippocampal function. Our point of departure was the observation that DG development is severely impaired in mouse embryos that are missing the gene coding for LEF1, a transcription factor that dimerizes with beta-catenin to mediate the transcriptional response to canonical Wnt signaling (*van Genderen et al., 1994*; *Galceran et al., 2000*). A similar, but milder, phenotype was reported for a knockout in the gene coding for LDL-receptor-related protein (LRP)6, one of two highly homologous co-receptors for canonical Wnt signaling (*Zhou et al., 2004*). As these mutants do not survive beyond birth (*Lrp6* KO) or weaning (*Lef1* KO), they are not useful for assessing the behavioral consequences of the reduction in DGCs.

Here we describe the developmental and behavioral consequences of reduced canonical Wnt signaling in cortical neuroglial progenitors following *Gfap-Cre*-mediated inactivation of a conditional allele of the *Wls* gene, which codes for the Wnt chaperone protein Wntless (*Bänziger et al., 2006*; *Bartscherer et al., 2006*). The resulting mice survive to adulthood and are healthy, but they lack ~90% of DGCs, including nearly all DGCs in the dorsal hippocampus. Other cell types within the hippocampus appear to be largely unaltered as determined by single nucleus RNA sequencing (snRNAseq), immunostaining, and morphometric analyses of CA3 pyramidal cells. Behavioral testing shows that the mutant mice have a modest performance decrement in cognitive tasks involving spatial learning and memory.

## Results

### Highly selective neuroanatomic defects in adult *Wls^{fl/-}*;*Gfap-Cre* mice

By comparing littermate *Wls^{fl/+}*;*Gfap-Cre* (phenotypically WT control) and *Wls^{fl/-}*;*Gfap-Cre* (mutant) mice, we observe that eliminating *Wls* in a subset of neural progenitors with *Gfap-Cre* has no effect on viability but is associated with (i) a ~ 25% reduction in body size and weight at early post-weaning ages that decreases to a ~ 5% reduction by six weeks of age and (ii) male infertility. The *Wls^{fl/-}*;*Gfap-Cre* and *Wls^{fl/+}*;*Gfap-Cre* progeny were derived from *Wls^{+/-}*;*Gfap-Cre/Gfap-Cre* male x *Wls^{fl/fl}* female parents, the standard genetic cross in all of the experiments that follow.

To explore the neuroanatomic consequences of eliminating *Wls* in a subset of neural progenitors with *Gfap-Cre*, we compared brains from 4- to 5-month-old *Wls^{fl/-}*;*Gfap-Cre* and *Wls^{fl/+}*;*Gfap-Cre* mice by (i) staining coronal sections with DAPI or cresyl violet to visualize cell bodies (*Figure 1A*;>10 mutant brains analyzed) and (ii) by micro-diffusion tensor imaging (uDTI) to visualize axon tracts (*Figure 1B*; three mutant brains analyzed). Examination of the principle cortices, tracts, and nuclei revealed only two visible alterations in *Wls^{fl/-}*;*Gfap-Cre* brains: (i) absence of the corpus callosum and (ii) a large reduction in the size of the dentate gyrus (DG). In some *Wls^{fl/-}*;*Gfap-Cre* brains the DG appears to be almost completely missing, whereas, in others, the most ventral (i.e., posterior) region of the hippocampus retains the DG at a reduced size (*Figure 1—figure supplement 1*).

Visualizing DG granule cells and mossy cells by immunostaining for Prox1 and calretinin, respectively, showed that DGCs are largely missing from *Wls^{fl/-}*;*Gfap-Cre* brains whereas DG mossy cells are retained (*Figure 1C*, left). Consistent with this pattern of cell body loss, visualizing mossy fibers, the axons of the DGCs that synapse onto CA3 pyramidal cells, by immunostaining for Calbindin showed a corresponding loss of mossy fibers in *Wls^{fl/-}*;*Gfap-Cre* brains (*Figure 1C*, right).

### Developmental basis of the *Wntless* conditional mutant phenotype

To explore the origin of the DG phenotype in *Wls^{fl/-}*;*Gfap-Cre* mice, we first mapped the spatiotemporal pattern of *Gfap-Cre* expression using a *R26-LSL-tdT-2A-H2BGFP* reporter in which a *LoxP-stop-LoxP* cassette (*LSL*) blocks expression of two fluorescent reporter proteins, a membrane tdTomato and a H2B-GFP fusion, both of which are under the control of a widely expressed *CAG* promoter at the *Rosa26* locus (*Wang et al., 2018*). This analysis revealed *Gfap-Cre* expression in the medial and dorsal cerebral cortex by embryonic day (E)14.5, including the cortical hem, an organizing center for hippocampal development that is located at the most medial edge of the developing cortex (*Figure 2A*; *Grove et al., 1998*). To assess the spatiotemporal pattern of beta-catenin signaling, we took advantage of the observation that in some tissues, such as CNS endothelial cells, expression of the beta-catenin partner LEF1 (also known as LEF/TCF), is induced by beta-catenin signaling as part of a presumptive positive feedback loop (*Wang et al., 2019*). Immunostaining for LEF1 shows the highest intensity in the cortical hem with similar levels in both *Wls^{fl/-}*;*Gfap-Cre* and *Wls^{fl/+}*;*Gfap-Cre* brains at E14.5 (*Figure 2A*, upper panels). However, at E15.5, LEF1 levels in the cortical hem are markedly lower in *Wls^{fl/-}*;*Gfap-Cre* compared to *Wls^{fl/+}*;*Gfap-Cre* brains (*Figure 2A*, lower panels). Despite the reduced LEF1 level, the cellular architecture of the *Wls^{fl/-}*;*Gfap-Cre* cortex appears normal, including the abundance and locations of Cajal–Retzius cells, which are important for normal DG development (*Figure 2—figure supplement 1*; *Hodge et al., 2013*). At E18.5, a time when the mature hippocampal architecture is recognizable, LEF1 accumulation is localized to the region of the future DG in control *Wls^{fl/+}*;*Gfap-Cre* brains (*Figure 2B*, upper panels), but the analogous territory in *Wls^{fl/-}*;*Gfap-Cre* brains contains fewer cells and shows greatly reduced LEF1 levels (*Figure 2B*, lower panels).

In many normal developmental contexts, as well as in the context of neoplasia, beta-catenin signaling drives cell proliferation, and therefore one likely explanation for a reduction in DGCs is decreased cell proliferation secondary to reduced beta-catenin signaling in the cortical hem and adjacent medial cortex. In support of this idea, the number of EdU-positive cells in the region that gives rise to the DG, marked by Prox1 immunostaining, was reduced ~5 fold in *Wls^{fl/-}*;*Gfap-Cre* brains compared to *Wls^{fl/+}*;*Gfap-Cre* brains at E18.5 (*Figure 2C and D*). By contrast, the number of EdU-positive neural progenitors in the adjacent hippocampal migratory stream showed less than a 2-fold difference. Immunostaining for cleaved Caspase-3 showed minimal cell death in both *Wls^{fl/-}*; *Gfap-Cre* and *Wls^{fl/+}*;*Gfap-Cre* brains at E17, E18.5, and P0 (data not shown). We conclude that the

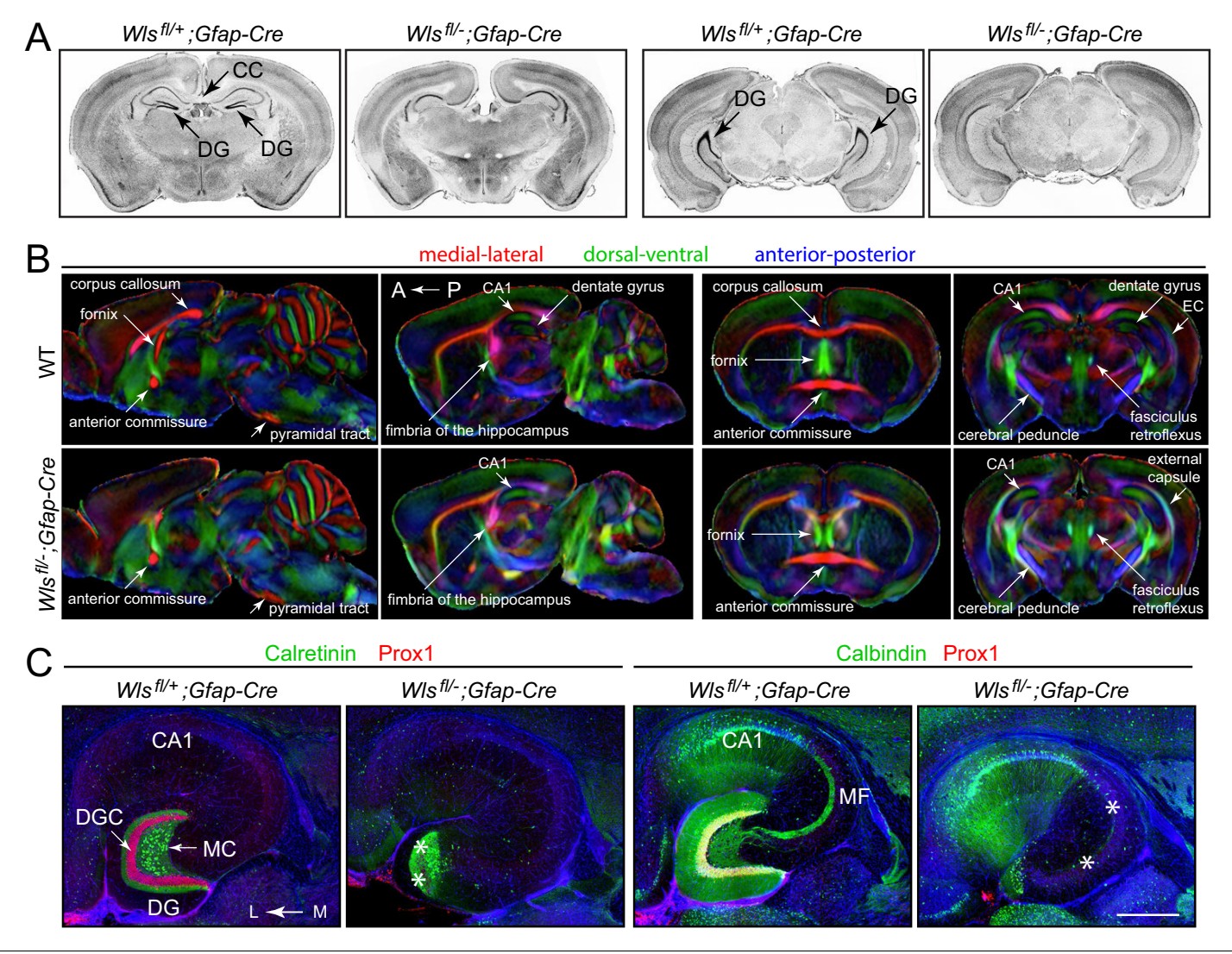

**Figure 1.** Hypoplasia of the DG and absence of the corpus callosum in adult *Wls^fl/-^;Gfap-Cre* mice. (**A**) Coronal brain sections from adult *Wls^fl/+^;Gfap-Cre* and *Wls^fl/-^;Gfap-Cre* mice at the level of the anterior/dorsal hippocampus (left pair of images) and posterior/ventral hippocampus (right pair of images). Additional coronal sections from these two mice and from a second *Wls^fl/-^;Gfap-Cre* mouse are shown in *Figure 1—figure supplement 1*. (**B**) µDTI images from 7month-old *Wls^fl/+^;Gfap-Cre* and *Wls^fl/-^;Gfap-Cre* brains. Each image represents an average of three brains of the indicated genotype. Left images, sagittal sections. Right images, coronal sections. (**C**) Horizontal sections through dorsal adult *Wls^fl/+^;Gfap-Cre* and *Wls^fl/-^;Gfap-Cre* hippocampi, immunostained with the indicated antibodies and counterstained with DAPI. Asterisks mark the locations of missing DGCs and mossy fibers. CC, corpus callosum. DG, dentate gyrus. DGC, dentate granule cells. EC, external capsule. MC, mossy cells. MF, mossy fibers. A, anterior. P, posterior. L, lateral. M, medial. Scale bar in C, 500 µm.

The online version of this article includes the following figure supplement(s) for figure 1:

**Figure supplement 1.** Serial coronal sections of adult *Wls^fl/+^;Gfap-Cre* and *Wls^fl/-^;Gfap-Cre* brains.

nearly complete absence of DGCs in the adult *Wls^fl/-^;Gfap-Cre* brain reflects a failure to produce these cells during development rather than production followed by loss.

## Differentially expressed genes in the *Wntless* conditional mutant hippocampus: bulk RNAseq

As a first step in assessing changes in cell composition and in patterns of gene expression, we performed RNAseq analysis on dissected hippocampi from 12-week-old male *Wls^fl/-^;Gfap-Cre* and *Wls^fl/+^;Gfap-Cre* mice, with two biological replicates per genotype. Based on analyses performed by

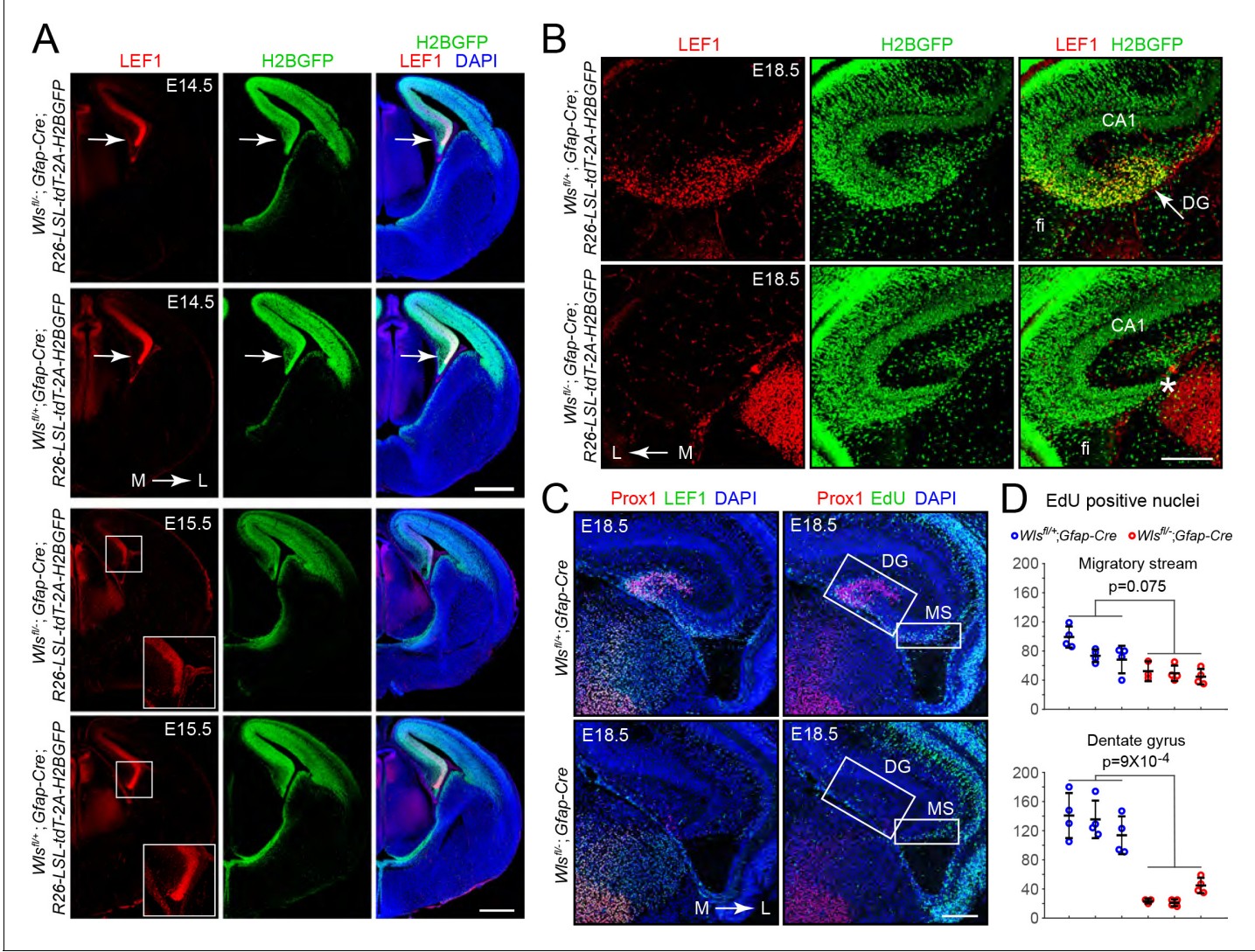

**Figure 2.** Hypo-proliferation of DG progenitors in *Wls^{fl/-};Gfap-Cre* fetuses. (**A**) Coronal sections of E14.5 brains (upper six images) and E15.5 brains (lower six images) from *Wls^{fl/+};Gfap-Cre;R26-LSL-tdT-2A-H2BGFP* and *Wls^{fl/-};Gfap-Cre;R26-LSL-tdT-2A-H2BGFP* fetuses showing (i) the territory of *Gfap-Cre* expression as indicated by the accumulation of the H2B-GFP reporter and (ii) the location and magnitude of beta-catenin signaling as indicated by the accumulation of LEF1. In the E14.5 images, the arrow points to the cortical hem. In the E15.5 images, the cortical hem region is boxed and enlarged at the lower right. (**B**) Coronal section and immunostaining as in (**A**) except at E18.5. Asterisk, location where the DG should be. (**C**) Horizontal sections of E18.5 *Wls^{fl/+};Gfap-Cre* and *Wls^{fl/-};Gfap-Cre* brains with developing DG pyramidal neurons visualized by Prox1 immunostaining, the response to beta-catenin signaling visualized by LEF1 immunostaining, and cell proliferation visualized by EdU labeling following an injection 2 hr prior to sacrifice. MS, migratory stream. (**D**) Quantification of EdU- positive nuclei in the developing migratory stream (upper) and the adjacent DG (lower; identified by Prox1 immunostaining) in E18.5 *Wls^{fl/+};Gfap-Cre* and *Wls^{fl/-};Gfap-Cre* brains. Each data point represents the counts from a confocal Z-plane of 15 μm thickness. Bars show mean ± S.D. Fi, fimbria. L, lateral. M, medial. Scale bar in (**A**), 500 μm. Scale bars in (**B**) and (**D**), 200 μm.
The online version of this article includes the following figure supplement(s) for figure 2:

**Figure supplement 1.** Reduced beta-catenin signaling in DG progenitors in *Wls^{fl/-};Gfap-Cre* fetuses at E15.5.

the Allen Brain Atlas and the Hipposeq projects (*Cembrowski et al., 2016*), transcripts specific to CA1, CA2, or CA3 pyramidal neurons, DG mossy cells (MCs), or DGCs were identified, and these are plotted as orange symbols in the scatterplots in *Figure 3A*. The RNAseq data revealed 60 transcripts with reduced abundance in the *Wls^{fl/-};Gfap-Cre* hippocampus that met the criteria of fold change (FC) >3 and false discovery rate (FDR) < 0.05. Strikingly, 55 of the 60 transcripts are specifically expressed or substantially enriched in DGCs (*Figure 3A*). Three of the 60 transcripts are expressed specifically in other hippocampal cell types (*Mafa*, expressed in CA3 pyramidal cells; *Gal*,

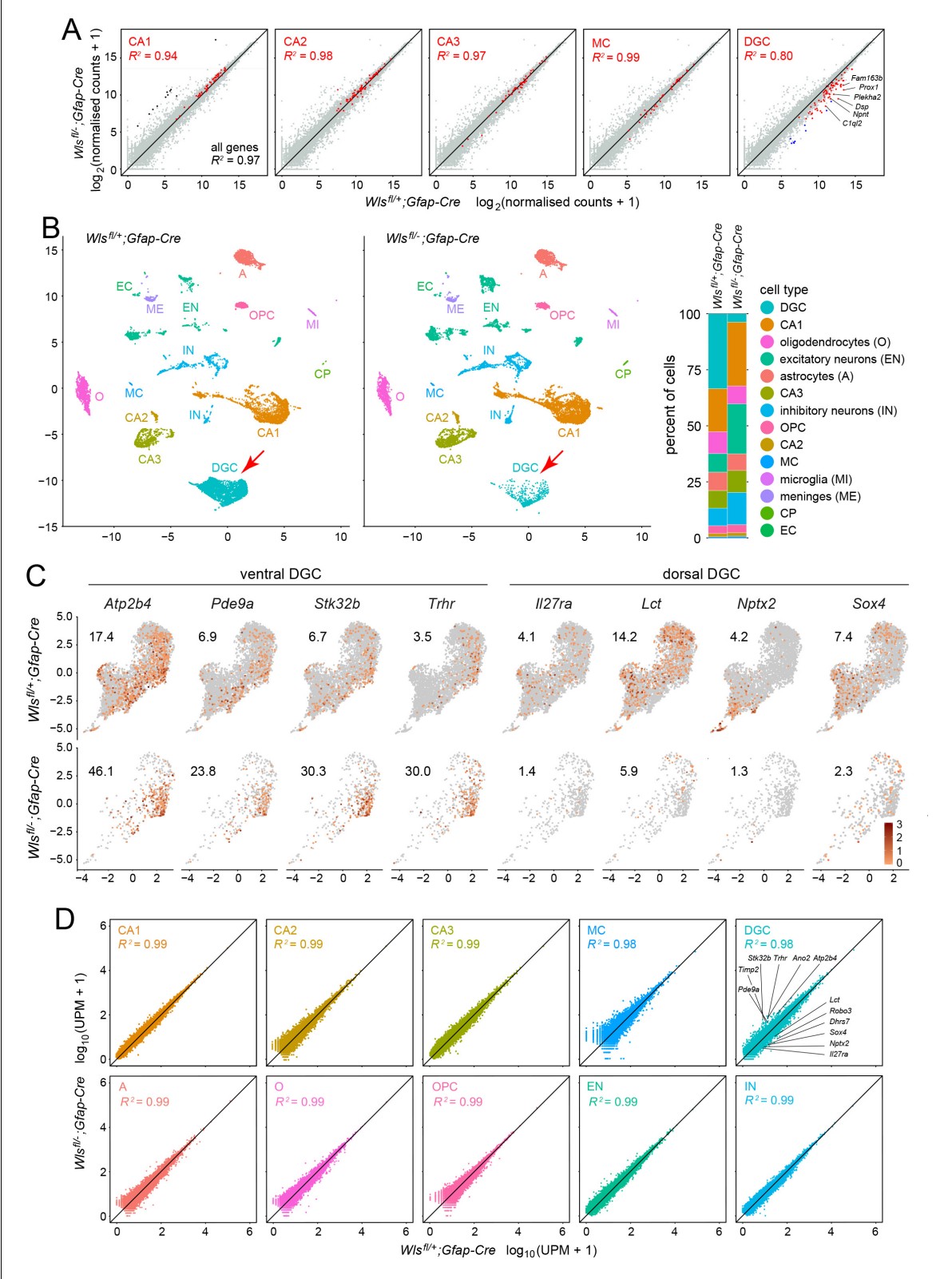

**Figure 3.** Bulk RNAseq and snRNAseq show loss of granule cells in the dorsal DG, with little change in other hippocampal cell types. (**A**) Scatterplots of normalized read counts from bulk RNAseq from ~12 week old *Wls*^*fl/+*^;*Gfap-Cre* and *Wls*^*fl/-*^;*Gfap-Cre* hippocampi. Gray symbols represent all transcripts. Red symbols represent transcripts previously reported to be enriched in each of the five indicated cell types (from left to right): CA1 pyramidal cells, CA2 pyramidal cells, CA3 pyramidal cells, mossy cells (MC), and dentate granule cells (DGC). Black symbols in the left-most scatter plot represent

*Figure 3 continued on next page*

Figure 3 continued

transcripts enriched in choroid plexus. Blue symbols in the right-most scatter plot represent transcripts enriched in DGCs based on the snRNAseq analysis reported here. In the CA2 scatter plot, the lone red data point that is well above the 45-degree line is expressed in both CA2 pyramidal neurons and the choroid plexus. By snRNAseq, its expression in CA2 pyramidal neurons is unaffected by genotype. (B) UMAP plot of cell clusters from snRNAseq of ~12 week old *Wls*^*fl/+*^*;Gfap-Cre* and *Wls*^*fl/-*^*;Gfap-Cre* hippocampi. Red arrow, DGC cluster. Right, percentile plots of the major hippocampal cell types in control and mutant snRNAseq datasets. (C) DGC-only UMAP plots showing normalized read counts for transcripts enriched in the ventral DG (left) and the dorsal DG (right) in *Wls*^*fl/-*^*;Gfap-Cre* vs. *Wls*^*fl/+*^*;Gfap-Cre* hippocampi. Numbers associated with each UMAP plot indicate the percentage of cells expressing the indicated gene.(D) Scatter plots comparing *Wls*^*fl/-*^*;Gfap-Cre* vs. *Wls*^*fl/+*^*;Gfap-Cre* transcripts for the ten most abundant hippocampal cell types. For each of the ten clusters, read counts were pooled and normalized. For the DGC cluster (upper right), the twelve transcripts shown in *Figure 3—figure supplement 2* are identified on the scatter plot: six transcripts are enriched in the ventral DG and reside above the 45-degree line, and six transcripts are enriched in the dorsal DG and reside below the 45-degree line.

The online version of this article includes the following figure supplement(s) for figure 3:

**Figure supplement 1.** Hippocampal cell-type clustering based on transcript abundances.

**Figure supplement 2.** Comparisons of normalized snRNAseq read counts for DGC transcripts enriched in the ventral DG or the dorsal DG in *Wls*^*fl/-*^*;Gfap-Cre* vs.*Wls*^*fl/+*^*;Gfap-Cre* hippocampi.

**Figure supplement 3.** An additional dimension of DGC subclusters in the DGC-only UMAP cluster.

**Figure supplement 4.** Comparisons of normalized snRNAseq read counts for twelve transcripts in DGCs of *Wls*^*fl/-*^*;Gfap-Cre* vs.*Wls*^*fl/+*^*;Gfap-Cre* hippocampi.

expressed in the mossy cells of the dentate hilus; and *Dsc3* expressed in excitatory neurons), and the two additional non-DG transcripts were barely detectable using snRNAseq (described below), and therefore their cell-type specificity is unclear. Nine additional transcripts with significantly reduced abundance in the *Wls*^*fl/-*^*;Gfap-Cre* hippocampus that met the FC and FDR criteria, and that had no assigned cell type in the Allen Brain Atlas or Hipposeq data sets, were subsequently found to be enriched in DGCs by snRNAseq (*Figure 3A*; blue dots in the right-most plot).

The bulk RNAseq analysis also revealed 34 transcripts that met the FC and FDR criteria for increased abundance in the *Wls*^*fl/-*^*;Gfap-Cre* hippocampus. Twenty-seven of these transcripts are specifically expressed in the choroid plexus (black dots; *Figure 3A*, left panel), and, therefore, their presence likely indicates choroid plexus contamination in the *Wls*^*fl/-*^*;Gfap-Cre* hippocampus samples during dissection. The remaining seven transcripts showed little or no signal in the snRNAseq data, and therefore their cell-type specificity is unclear.

Close inspection of the scatterplots in *Figure 3A* shows that cell-type-specific transcripts for mossy cells and CA1, CA2, and CA3 pyramidal cells reside, on average, just above the 45-degree line, whereas the cell-type-specific transcripts for DGs reside, on average, substantially below the 45-degree line. These data are consistent with a substantial reduction in the number of DGs in the *Wls*^*fl/-*^*;Gfap-Cre* hippocampus and a resulting small increase in the relative representation of all other cell types, but with little or no change in the patterns of gene expression.

## Differentially expressed genes in the *Wntless* conditional mutant hippocampus: snRNAseq

Cell-type-specific changes in the transcriptome can be difficult to detect using whole-tissue RNAseq, especially in tissues like the hippocampus that consist of complex mixtures of cells. Therefore, to more precisely explore transcriptional changes in the *Wls*^*fl/-*^*;Gfap-Cre* hippocampus, we performed single nucleus (sn)RNAseq on dissected hippocampi from 12 week old male *Wls*^*fl/-*^*;Gfap-Cre* and *Wls*^*fl/+*^*;Gfap-Cre* mice using the 10x Genomics Chromium platform with V3 chemistry. Transcripts were sequenced from 15,573 nuclei from *Wls*^*fl/-*^*;Gfap-Cre* hippocampi and 13,523 nuclei from *Wls*^*fl/+*^*;Gfap-Cre* hippocampi. The resulting Uniform Manifold Approximation and Projection (UMAP) clusters correspond to all of the major cell types in the hippocampus (*Figure 3B* and *Figure 3—figure supplement 1*). All but one of the UMAP clusters appear nearly identical between *Wls*^*fl/-*^*;Gfap-Cre* and *Wls*^*fl/+*^*;Gfap-Cre* datasets. The exception is the DGC cluster, which shows a significant reduction in cell number in the *Wls*^*fl/-*^*;Gfap-Cre* sample (red arrows in *Figure 3B*), consistent with the histologic and the whole-tissue RNAseq data (*Figure 1—figure supplement 1*, and *Figure 3A*).

As summarized in the percentile plot in *Figure 3B*, DGCs comprise 32% of the *Wls*^*fl/+*^*;Gfap-Cre* nuclei (4306 of 13,523) but only 3.5% of the *Wls*^*fl/-*^*;Gfap-Cre* nuclei (548 of 15,573). All other neuronal cell types show increased cell numbers in the *Wls*^*fl/-*^*;Gfap-Cre* data set, with the greatest

increases seen for (1) inhibitory neurons, which comprise 9% of the *Wls$^{fl/+}$;Gfap-Cre* nuclei and 15.5% of the *Wls$^{fl/-}$;Gfap-Cre* nuclei, and (2) excitatory neurons other than the CA1, CA2, CA3 pyramidal cells, which comprise 8% of the *Wls$^{fl/+}$;Gfap-Cre* nuclei and 21% of the *Wls$^{fl/-}$;Gfap-Cre* nuclei. Part of the increase in the abundance of non-DGCs in the *Wls$^{fl/-}$;Gfap-Cre* sample arises from the reduction in the abundance of DGCs, an effect that should produce a 1.3-fold increase in relative abundance for all non-DGCs. Abundance changes differing from 1.3-fold might arise, at least in part, from variation between experiments in the yield of different classes of nuclei. More interesting is the possibility that, reduced beta-catenin signaling in the developing *Wls$^{fl/-}$;Gfap-Cre* hippocampus either directly or indirectly increases the numbers of inhibitory neurons and non-pyramidal excitatory neurons. As an example of an indirect mechanism, the production or survival of these neurons might be enhanced by a reduction in DGCs. Whatever the mechanism(s) responsible for the greater numbers of inhibitory neurons and non-pyramidal excitatory neurons in the *Wls$^{fl/-}$;Gfap-Cre* hippocampus, the snRNAseq data indicate that their intrinsic transcriptional programs are very similar in mutant and control hippocampi.

## Anatomic and transcriptome diversity among granule cells

DGC nuclei in the control and mutant snRNAseq data sets differ not only in abundance, but also in their locations within the DGC cluster in the UMAP plot, with *Wls$^{fl/-}$;Gfap-Cre* nuclei mainly occupying the lower part of the cluster (*Figure 3B*). This pattern suggests that the upper and lower parts of the DGC cluster correspond, respectively, to the dorsal (i.e. anterior) hippocampus and the ventral (i.e. posterior) hippocampus. To explore this observation in greater detail, new UMAP plots were generated for *Wls$^{fl/-}$;Gfap-Cre* and *Wls$^{fl/+}$;Gfap-Cre* datasets using only DGC nuclei, and the patterns and abundances of individual transcripts were plotted, together with the percentage of DGC cells expressing each gene (*Figure 3C*). [The DGC-only UMAP cluster closely resembles the whole hippocampus DGC UMAP cluster, except with a ~ 60 degree counter-clockwise rotation].

For 12 DGC-expressed genes, including the eight genes plotted in *Figure 3C*, normalized RNA-seq read counts were calculated for all DGCs with any reads for the indicated gene, and the RNA in situ hybridization (ISH) pattern with the corresponding probe was assessed in parasagittal sections of adult mouse brain, as shown in the Allen Brain Atlas (*Figure 3—figure supplement 2*). The ISH images reveal differences in transcript abundances in dorsal vs. ventral hippocampus, as well as the extent of enrichment within the DG. For example, *Lct* transcripts are more abundant in the dorsal hippocampus and *Trhr* transcripts are more abundant in the ventral hippocampus (*Figure 3—figure supplement 2*). These conclusions are in accord with those of *Cembrowski et al., 2016* who performed RNAseq on pools of manually dissected hippocampal neurons from micro-dissected tissue. Combining the information from the ISH and UMAP patterns shows that, within the UMAP DGC cluster in *Figure 3C*, an extended arc of cells residing on the lower right side of the cluster corresponds to the ventral hippocampus and the remainder of the DGC cluster corresponds to the dorsal hippocampus.

To look for additional gene expression patterns within the DGC cells cluster, we used the Monocle 3 'find_gene_modules' algorithm. By visual inspection, at least six distinct expression modules are present within the DGC cluster, with module one largely overlapping with the ventral DG. Two of these modules (modules 7 and 11) define mutually exclusive sub-domains within the DGC cluster that are orthogonal in UMAP space to the dorsal/ventral sub-domains (*Figure 3—figure supplement 3*). As inspection of the ISH data in the Allen Brain Atlas did not reveal large-scale patterns within the DG for transcripts enriched in modules 7 or 11, the anatomic and functional correlates of this subdivision remain to be determined.

The quantification of snRNAseq read counts per expressing cell for the six dorsal-enriched and the six ventral-enriched transcripts shown in *Figure 3—figure supplement 2* indicates that, despite their reduced numbers, the DGCs that remain in the *Wls$^{fl/-}$;Gfap-Cre* hippocampus express ventral and dorsal markers at very nearly the same levels as their counterparts in the *Wls$^{fl/+}$;Gfap-Cre* hippocampus. An extension of this quantification to other classes of DGC transcripts shows that this pattern holds generally (*Figure 3—figure supplement 4*).

To systematically search for transcripts that were differentially expressed on a transcripts-per-cell basis between *Wls$^{fl/-}$;Gfap-Cre* and *Wls$^{fl/+}$;Gfap-Cre* in the snRNAseq data, we applied a regression analysis to individual hippocampal cell types using the Monocle-3 R package. The cell types tested corresponded to the most abundant classes shown in *Figure 3B*: CA1, CA2, and CA3 pyramidal

neurons, MCs, DGCs, astrocytes, oligodendrocytes, oligodendrocyte precursors, other excitatory neurons (several clusters identified by expression of *Slc17a7*, *Nrn1*, *Nrgn*, *Fh12*, and *Neurod2* transcripts), and inhibitory neurons (several clusters identified by expression of *Gad1*, *Gad2*, *Kcnip1*, *Erbb4*, *Rbms3*, and *Kcnmb2* transcripts). Transcripts were identified as differentially expressed if they showed a FC >2 with a q-value <0.05. Remarkably, only two transcripts fulfilled these criteria: *Trps1* and *Cntnap5a*, both in a subset of excitatory neurons. Among excitatory neurons with non-zero read counts, *Trps1* had a mean read count of 4.5 in the mutant and 1.6 in the control with p-value = $2\times10^{-16}$, and *Cntnap5a* had a mean read count of 6.9 in the mutant and 2.9 in the control with p-value = $2\times10^{-16}$. The high similarity between *Wls*$^{fl/-}$*;Gfap-Cre* and *Wls*$^{fl/+}$*;Gfap-Cre* transcriptomes for each of these ten hippocampal cell types is apparent in scatterplots of the snRNAseq data that show UMI per million (UPM) for each expressed gene (*Figure 3D*). MC and DGC scatter plots have $R^2$ = 0.98, and the other eight scatter plots have $R^2$ = 0.99. The marginally lower correlation coefficient for mutant vs. control DGC transcripts likely reflects the different representation of dorsal-enriched and ventral-enriched transcripts, as seen by the locations of these data points in *Figure 3D*.

In sum, bulk RNAseq and snRNAseq reveal a large loss of DGCs in *Wls*$^{fl/-}$*;Gfap-Cre* mice but almost no effects on the transcriptomes of these or any other hippocampal cell types.

## CA3 pyramidal cell synapses in the *Wntless* conditional mutant hippocampus

DGC axons, which constitute the mossy fiber bundle, project to the CA3 region where they form synapses with both pyramidal cells and inhibitory interneurons (*Jaffe and Gutiérrez, 2007*). DGC-pyramidal cell synapses are composed of *en passant* presynaptic boutons from DGC axons and complex postsynaptic spines on the proximal dendrites of CA3 pyramidal cells (*Rollenhagen and Lübke, 2006*).

To determine whether the nearly complete loss of DGCs in the dorsal *Wls*$^{fl/-}$*;Gfap-Cre* hippocampus alters the structure of CA3 pyramidal cells, Golgi staining was used to reconstruct individual CA3 pyramidal cells in the dorsal half of the hippocampus and to quantify the density and type of dendritic spines. Eighteen CA3 pyramidal cells from area CA3b (encompassing the highly curved region of the hippocampus) were fully reconstructed from young adult brains, nine from *Wls*$^{fl/-}$*;Gfap-Cre* and nine from *Wls*$^{fl/+}$*;Gfap-Cre*. By visual inspection, *Wls*$^{fl/-}$*;Gfap-Cre* CA3 pyramidal cell morphologies appear unaffected, and there were no significant differences in the lengths of apical or basal dendrites between genotypes (*Figure 4A and B*).

Among the 18 reconstructed cells, dendritic spines were classified as either complex (thorny excrescences) or simple based on the criteria of *Sorra and Harris, 2000* and *Gonzales et al., 2001*. The mean number of spines per cell did not differ significantly between genotypes, and separate comparisons of complex and simple spines along apical and basal dendrites showed no significant differences in mean spine number in any of the four pairwise comparisons between genotypes (*Figure 4C*, upper panels). Analogous comparisons performed for spine density per unit length of dendrite showed modest and statistically significant reductions in the mean density of complex spines on apical dendrites and of simple spines on basal dendrites in the *Wls*$^{fl/-}$*;Gfap-Cre* cells (*Figure 4C*, lower panels).

To investigate the structure of complex spines on CA3 pyramidal cells, the length of each complex spine along the dendrite was measured and extracted using the Spine Detail function in Neurolucida Explorer. For complex spine length, there was no significant difference between genotypes (Wilcoxon rank sum = 2.24, p=0.67; *Figure 4D*). In view of the modestly lower mean density of complex spines on apical dendrites, we determined whether the total coverage of dendrites with complex spines differed between the two genotypes. This analysis shows that there is significantly higher dendritic coverage with complex spines in the controls (Wilcoxon Rank Sum = 110, p=0.03; *Figure 4E*).

As a second approach to comparing CA3 pyramidal spine structure and density between *Wls*$^{fl/-}$*;Gfap-Cre* and *Wls*$^{fl/+}$*;Gfap-Cre* hippocampi, an unbiased stereological approach was used to count and classify complex dendritic spines in CA3 (*Supplementary files 1–3*). Based on the observations of *Gonzales et al., 2001*, *Tsamis et al., 2010*, and *Amaral, 1978*, complex spines were classified into five morphological subtypes: basic, big/prototypical, long, tall, and thin. To sample the entire dorsal-ventral extent of the hippocampus from one hemisphere for each genotype, 25 section of a

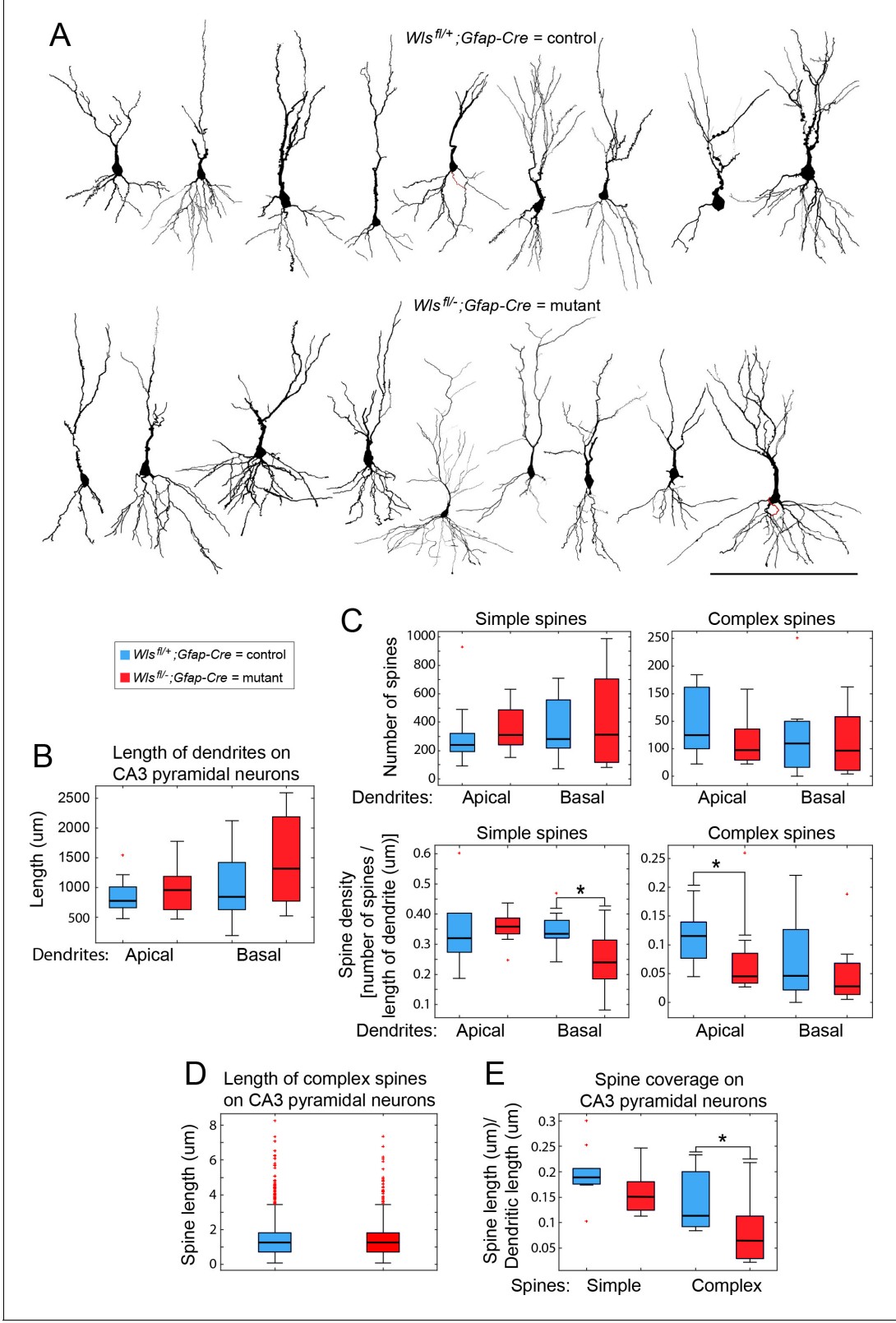

**Figure 4.** Morphological analyses of CA3 pyramidal cells and their synapses. (**A**) CA3 pyramidal cells in dorsal hippocampus, reconstructed from Golgi-stained 2–3-month-old *Wls^{fl/+};Gfap-Cre* and *Wls^{fl/-};Gfap-Cre* brains. Apical is up and basal is down. (**B**) Lengths of apical and basal dendrites among the 18 reconstructed CA3 pyramidal cells. (**C**) Numbers and densities of complex and simple dendritic spines on the 18 reconstructed CA3 pyramidal cells. (**D,E**) Length of complex dendritic spines and spine coverage by simple and complex spines on the 18 reconstructed CA3 pyramidal cells. Box plots

*Figure 4 continued on next page*

*Figure 4 continued*

show the median and 25th to 75th percentiles, whiskers include all data points not considered outliers, and individual outliers are shown. Scale bar in (A), 200 µm. * represents p<0.05.

*Wls^fl/+^;Gfap-Cre* hippocampus and 26 sections of a *Wls^fl/+^;Gfap-Cre* hippocampus were analyzed (**Supplementary file 1**). Both showed complex spines throughout the dorso-ventral extent of the hippocampus.

The total number of complex spines counted in the *Wls^fl/-^;Gfap-Cre* CA3 was ~65% of the number counted in the *Wls^fl/+^;Gfap-Cre* CA3. A one-sided t-test between the counted spines showed that the *Wls^fl/-^;Gfap-Cre* CA3 had significantly fewer sampled sites with complex spines compared to the *Wls^fl/+^;Gfap-Cre* CA3 [329/1433 (22%) vs. 369/1415 (26%); t(49) = 2.43, p=0.01]. Our interpretation of this comparison assumes that the efficiency of Golgi staining was equivalent between the two brains, which were processed in parallel. Among the subtypes of complex spines, the *Wls^fl/-^;Gfap-Cre* CA3 had significantly fewer basic spines (Wilcoxon rank sum = 523.5, p<0.01) and big/prototypical spines (Wilcoxon rank sum = 505, p<0.01) but the number of tall spines (Wilcoxon rank sum = 397.5, p=0.09), long spines (Wilcoxon rank sum = 368.5, p=0.21,) and thin spines (Wilcoxon rank sum = 381, p=0.15) were not significantly different (**Supplementary file 3**).

Based on these analyses, we conclude that *Wls^fl/-^;Gfap-Cre* CA3 pyramidal cells have normal morphologies but they exhibit a modest reduction in the number of synaptic spines – in particular, a modest reduction in the density of and coverage with complex spines on apical dendrites – an effect that may be secondary to the large reduction in DGC inputs.

## Phenotype of the *Wntless Gfap-Cre* conditional mutant in baseline behavioral tasks

The relatively simple cellular and molecular phenotype in the *Wls^fl/-^;Gfap-Cre* hippocampus – a large reduction in DGCs with minimal effects on other cell types – recommends this mutant for behavioral phenotyping. In the paragraphs that follow, we describe the results from a battery of tests in which we compared male and female *Wls^fl/-^;Gfap-Cre* and *Wls^fl/+^;Gfap-Cre* mice at 10–20 weeks of age using 14–20 mice from each genotype (**Figures 5**, **6**, **7**). During all behavioral testing, the experimenters were blind to the genotype. No systematic differences were seen between the sexes, and the male and female data were therefore pooled for the principal analysis. Examples of data separated by sex are shown in **Figure 5—figure supplements 1–3**.

As a first step in behavioral phenotyping, *Wls^fl/-^;Gfap-Cre* mice were tested for baseline neurologic function: climbing on a vertical screen, spontaneous and elicited grooming, turning on parallel bars, visual placing, negative geotaxis, and the suspension test. With the exception of negative geotaxis, there was no statistically significant effect of genotype on any of these tests (**Supplementary file 4**). Spontaneous locomotion was assessed in the open field test, motor learning was assessed in the rotarod test, anxiety-related behavior was assessed in the elevated plus maze, and basic cognitive performance was assessed in the Y-maze spatial recognition test. There was no effect of genotype on performance in any of these tests (**Figure 5—figure supplement 4**; p>0.05 in all cases).

## Phenotype of the *Wntless Gfap-Cre* conditional mutant in complex cognitive tasks, including spatial learning

To determine whether *Wls^fl/-^;Gfap-Cre* mice are impaired in more complex cognitive tasks, *Wls^fl/-^;Gfap-Cre* and *Wls^fl/+^;Gfap-Cre* mice were tested in the Barnes maze, the trace fear-conditioning task, and the Morris water maze. All mice learned the location of the escape hole in the Barnes maze, as indicated by a significant decrease in latency with training (**Figure 5A**, F(2.4, 84.5)=21.65, p=10^-8^). In a probe trial to determine if mice remembered the location of the escape hole 48 hr after training, the latency showed no statistically significant difference between genotypes (**Figure 5A**, t(35) = 0.81, p>0.05). In the trace fear-conditioning task, mice of both genotypes learned at a similar rate and increased freezing behavior in response to shock administration (**Figure 5B**, F(4.1,134.8) = 92.71, p<0.0001). Twenty-four hours after training, the two genotypes showed similar freezing responses in response to the context (**Figure 5B**, t(33) = 0.85, p>0.05) and

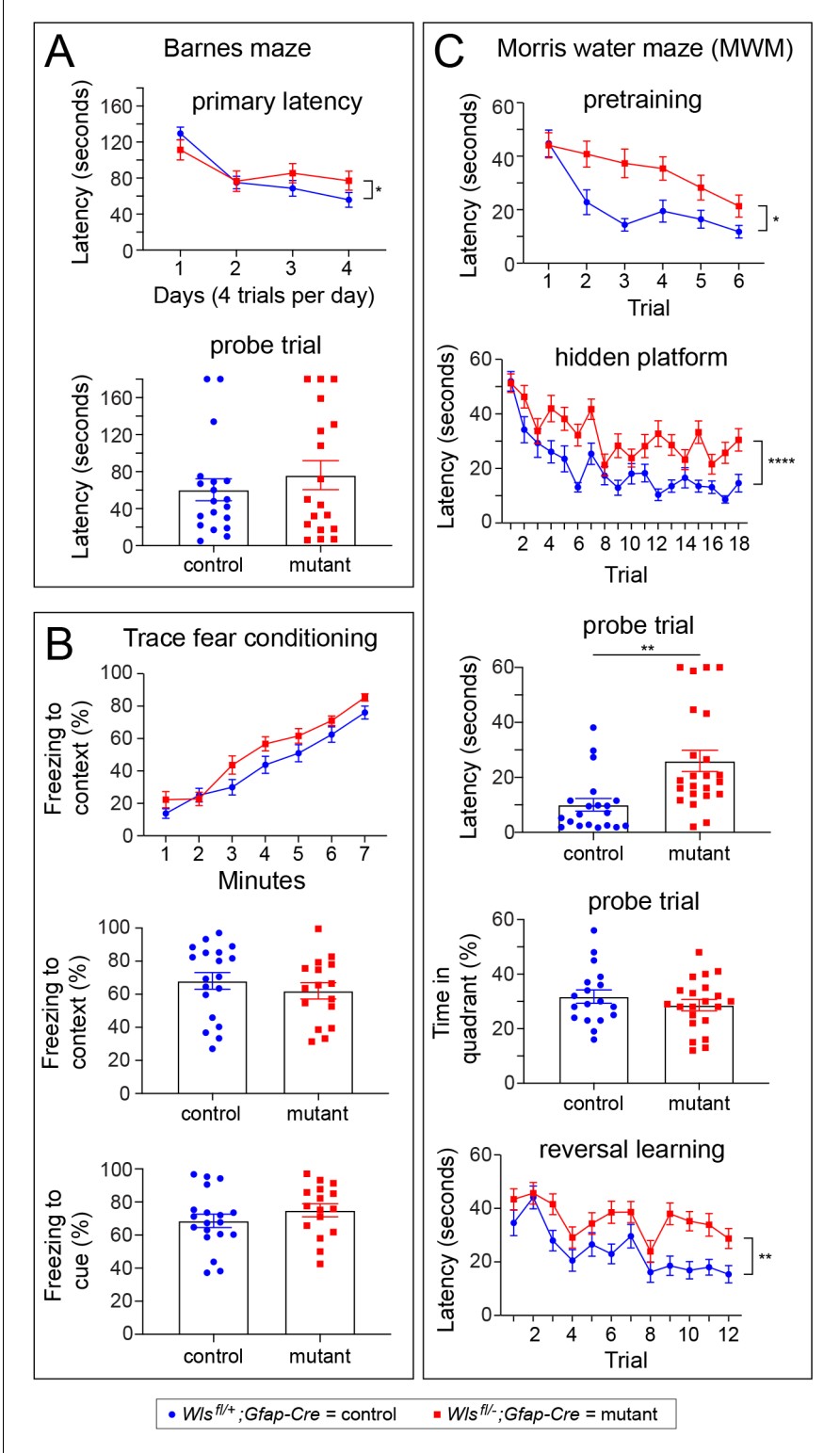

**Figure 5.** *Wls^{fl/-};Gfap-Cre* mice exhibit defects in the Morris water maze, but not in the Barnes maze or in a trace fear-conditioning task. Performance was assessed by comparing 10–20 week old *Wls^{fl/+};Gfap-Cre* and *Wls^{fl/-};Gfap-Cre* mice. Data were from approximately equal numbers of males and females for each genotype. See *Figure 5—figure supplements 1–3* for the number of males and females for each test and the results separated by sex. For plots showing the latencies for each trial, the vertical bracket indicates the statistical significance for the last trial. (**A**) Mutant mice exhibited marginally longer latency on day 4 of training in the Barnes maze (p<0.05; **A**), but no

*Figure 5 continued on next page*

*Figure 5 continued*

differences between control and mutant mice were found during a probe trial 48 hr later. (**B**) Freezing following trace fear conditioning showed no statistically significant difference between control and mutant mice during training (top panel) or in response to the context or cue 24 hr after training (bottom two panels). (**C**) In the MWM pretraining period with a visible platform, mutant mice showed a significantly greater latency than controls on trials 2–6 (p<0.05). During hidden platform training, an overall significant effect of trial was found (p<0.0001) as well as a significant increase in latency among mutant mice (p<0.001). The latency to approach the platform location was longer for mutant mice than controls during a probe trial 24 hr after hidden platform training (p<0.001). During the probe trial, no differences between mutant and control mice were found in percent time spent in the quadrant that previously contained the platform. In a reversal learning task following the probe trial, mutant mice showed a greater latency compared to controls (p<0.01) and there was an overall improvement in performance with successive trials for both genotypes (p<0.0001). *p<0.05, **p<0.01, ***p<0.001, ****p<0.0001, n = 16–21 mice per group. The graphs show mean +/- SEM.

The online version of this article includes the following figure supplement(s) for figure 5:

**Figure supplement 1.** No sex differences between adult *Wls*$^{fl/+}$*;Gfap-Cre* and *Wls*$^{fl/-}$*;Gfap-Cre* mice in basic behavioral tasks.

**Figure supplement 2.** Minimal sex differences between adult *Wls*$^{fl/+}$*;Gfap-Cre* and *Wls*$^{fl/-}$*;Gfap-Cre* mice in the Barnes maze, trace fear conditioning, and Morris water maze.

**Figure supplement 3.** Adult *Wls*$^{fl/-}$*;Gfap-Cre* mice exhibit reduced path efficiency in both the pretraining trials and in the probe trial in the Morris water maze.

**Figure supplement 4.** No difference between adult *Wls*$^{fl/+}$*;Gfap-Cre* and *Wls*$^{fl/-}$*;Gfap-Cre* mice in basic behavioral tasks.

the cue associated with the shock (*Figure 5B*, t(33) = 0.1.1, p>0.05). These data indicate that *Wls*$^{fl/-}$*; Gfap-Cre* mice are not impaired in fear learning or fear memory, and that they can learn and remember to discriminate between marked places as tested in the Barnes maze.

Locating the hidden platform in the Morris water maze (MWM) is a complex spatial learning and memory task. During pretraining, mice of both genotypes learned to locate a visible platform (*Figure 5C*, F(3.87, 158.70)=12.59, p<0.0001), although *Wls*$^{fl/-}$*;Gfap-Cre* mice exhibited a longer latency, as indicated by a significant genotype x trial effect (*Figure 5C*, F(5 and 20)=2.32, p<0.05). *Wls*$^{fl/-}$*;Gfap-Cre* mice also exhibited a statistically non-significant trend toward reduced path efficiency (*Figure 5—figure supplement 3A*). Importantly, *Wls*$^{fl/-}$*;Gfap-Cre* and *Wls*$^{fl/+}$*;Gfap-Cre* mice swam at similar speeds throughout the test (F(1,41) = 1.55, p>0.05, data not shown). During the hidden platform test for spatial learning, mice were placed in the MWM for six trials per day over three days. On the fourth day, they were given 60 s to locate the hidden platform. There was a significant decrease in latency in successive trials, indicating that mice of both genotypes could learn the task (*Figure 5C*; F(11.22, 460)=13.21, p<0.0001). However, compared to control mice, *Wls*$^{fl/-}$*;Gfap-Cre* mice displayed a longer latency to locate the hidden platform, suggestive of slower learning (F(1,41) =28.63, p<0.001). In the probe trial, one day after hidden platform training, *Wls*$^{fl/-}$*;Gfap-Cre* mice also displayed a significantly longer latency to locate the platform compared to control mice (*Figure 5C*, t(41) = 3.41, p<0.01) and a reduced path efficiency (*Figure 5—figure supplement 3B*; t(41) = 3.27, p<0.01). *Wls*$^{fl/-}$*;Gfap-Cre* and *Wls*$^{fl/+}$*;Gfap-Cre* mice spent a similar percent of time during the probe trial in the quadrant of the MWM in which the platform was located, indicating that mice of both genotypes had learned the general platform location (*Figure 5C*; t(37) = 0.99, p>0.05).

To determine if there was a difference in cognitive flexibility between *Wls*$^{fl/-}$*;Gfap-Cre* and *Wls*$^{fl/+}$*; Gfap-Cre* mice, a reversal test was conducted in which each mouse was given eight trials to learn a new location for the escape platform. There was an overall downward trend in latency in successive trials, indicating that mice of both genotypes could learn the new platform location (*Figure 5C*; F(7.76, 318.0)=9.73, p<0.0001). However, the latency decrease was more modest among *Wls*$^{fl/-}$*;Gfap-Cre* mice, implying that they were slower to learn the new platform location (*Figure 5C*; F(1,41) = 11.49, p<0.01). The equivalence between mutant and control mice in the time spent in the quadrant with the hidden platform suggests that the spatial learning deficit in *Wls*$^{fl/-}$*;Gfap-Cre* mice could be

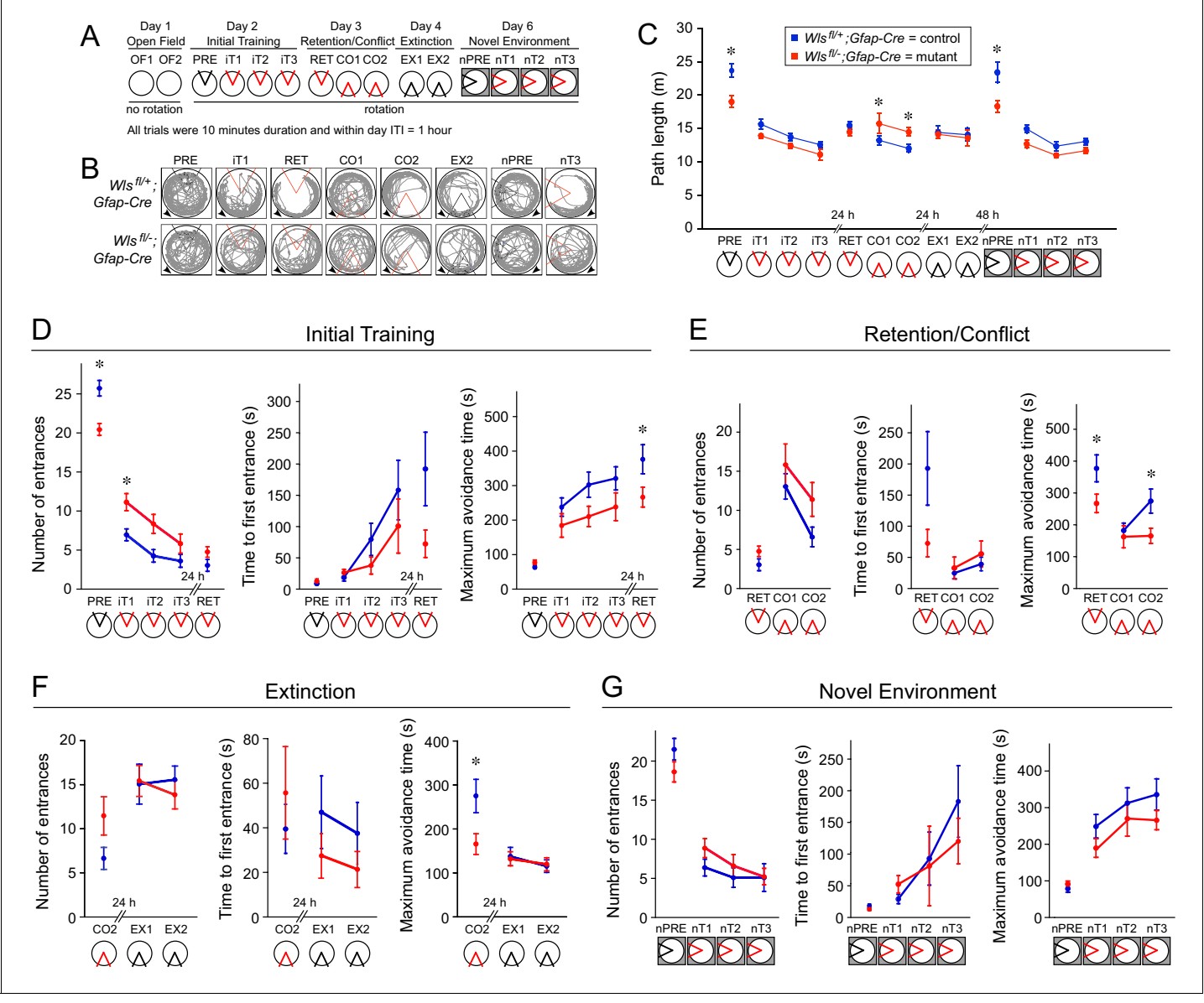

**Figure 6.** *Wls^fl/-^;Gfap-Cre* mice exhibit mild defects in active avoidance learning tasks. Fourteen *Wls^fl/-^;Gfap-Cre* and fifteen *Wls^fl/+^;Gfap-Cre* mice were tested. (**A**) Overview of the behavioral protocol for experiments on the rotating circular arena. A red 60° sector indicates the location of the shock zone. A black 60° sector indicates that the shock was turned off and entrances into this region were counted. The arena rotated at 1 rpm from day two onward. On day 6, the new arena was in a different location (indicated by the dark box). (**B**) Examples of active place avoidance behavior (shown by the mouse's trajectory in the room-frame), documented across the rotating arena protocol. Arrowheads show the direction of 1 rpm rotation. A red sector indicates the location of the 60° active shock zone, and a black sector indicates the zone location when the shock was turned off. (**C**) Path length measured across the rotating arena protocol. (**D**) Initial active place avoidance learning. Left, number of entrances (errors). Center, time to first enter the shock zone. Right, maximum time the shock zone was avoided. By all three measures of place avoidance, *Wls^fl/-^;Gfap-Cre* mice perform more poorly than the *Wls^fl/+^;Gfap-Cre* littermates. (**E**) Conflict in active place avoidance learning. Left, center, and right are as described for (**D**). (**F**) Extinction of the conditioned place avoidance. Left, center, and right are as described for (**D**). The mice partially extinguished their previously learned avoidance. (**G**) Subsequent active place avoidance learning in a novel environment. Left, center, and right are as described for (**D**). Bars indicate the mean ± SEM; * indicates $p < 0.05$ for comparisons between genotypes.

due, at least in part, to a defect in learning the location of the escape platform at high spatial resolution.

In sum, *Wls^fl/-^;Gfap-Cre* mice showed modestly impaired performance on all aspects of the MWM task: pretraining with a visible platform, training with a hidden platform, and reversal training with a

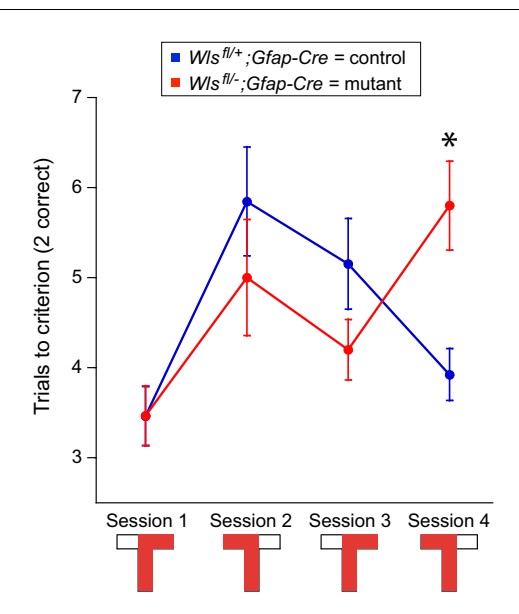

**Figure 7.** *Wls^fl/-;Gfap-Cre* mice perform more poorly in the last trial in an alternating T-maze task. Fourteen *Wls^fl/-;Gfap-Cre* and 15 *Wls^fl/+;Gfap-Cre* mice were tested in the alternating T-maze task. The graph shows mean +/- SEM.

hidden platform, consistent with an underlying performance deficit in the ability to escape from the water maze.

## Phenotype of the *Wntless Gfap-Cre* conditional mutant in an active avoidance task: initial assessment

As a second and independent measure of complex spatial learning and memory, *Wls^fl/-;Gfap-Cre* and *Wls^fl/+;Gfap-Cre* mice were tested in a six-day protocol to evaluate open field exploration and active place avoidance learning (*Figure 6*; *Cimadevilla et al., 2001*). This protocol is especially sensitive to hippocampal dysfunction, as it requires the mouse to selectively use stationary room-based information to avoid a shock zone and ignore rotating arena-based information (*Kubík and Fenton, 2005*). Mice were individually tested in a circular arena that was either stationary or rotating at one r.p.m. Within the arena, a foot shock was programmed within a fixed 60° sector, and this shock zone was either relocated to a new fixed location or deactivated for different tests. To provide novel visual cues, the apparatus was moved between rooms. As shown schematically in *Figure 6A*, the protocol consisted of (i) exploration in a stationary arena with no foot shock on day 1 (OF1 and OF2), (ii) initial training in a rotating arena on day 2, in which one pretrial with the shock turned off (PRE) was followed by three trials with the shock turned on (iT1-iT3), (iii) retention/conflict trials on day three in which a retention trial (RET) was followed by two trials with the shock zone relocated 180° (CO1 and CO2), (iv) two extinction trials on day four in which the shock zone was turned off (EX1 and EX2), and (v) four trials within a novel environment (a different apparatus in a different location, indicated in the schematic by a gray surround) on day six in which one pretrial with the shock turned off (nPRE) was followed by three trails with the shock turned on (nT1-nT3). Automated video-tracking software was used to track each mouse, and then compute the following values for the 60° shock zone: number of entrances, time to first entrance, and maximum avoidance time. *Figure 6B* shows the tracks of a representative mouse from each genotype during a subset of the trials. Throughout *Figure 6*, the 60° shock zone is colored red if the shock is turned on and black if the shock is turned off.

On day 1, mice from both genotypes explored the stationary open field similarly during the two trials (OF1 and OF2), with reduced exploration during the second trial. The distance walked (average = 13.9 m) characterized the open field behavior, and showed no effect of genotype ($F_{(1,26)}$ = 1.03, p=0.32) and no genotype x trial interaction ($F_{(1,52.9)}$ = 0.65, p=0.4), but a significant effect of trial ($F_{(1,52.8)}$ = 32.27, p=$10^{-7}$).

Pretraining on the rotating arena (PRE) is effectively a second open field test, and, during this test, the mutant mice walked less than the controls ($F_{(1,25)}$ = 10.6, p=0.003) (*Figure 6C*), thereby reducing the number of entrances into the 60° sector in the absence of any conditioning (*Figure 6D*, left panel). This difference prompted us to examine locomotion across the entire 6 day protocol (*Figure 6C*), since any difference in locomotion could contribute to a difference in active avoidance. This analysis showed that path length did not differ significantly between the genotypes across initial training when the mice avoided a shock ($F_{(1,35.1)}$ = 0.005, p=0.9) (iT1-iT3), although it decreased with training ($F_{(2,33.2)}$ = 15.64, iT1 >iT2=iT3; p=$10^{-5}$; interaction: $F_{(2,33.2)}$ = 0.66, p=0.5). Path length also did not significantly differ between genotypes on the retention test ($F_{(1,26.4)}$ = 0.55. p=0.5) (RET). When the shock zone was relocated to the opposite side of the arena in the two conflict trials (CO1 and CO2), the mutant mice walked ~20% more than the controls (genotype: $F_{(1,27)}$ = 8.88. p=0.006; trials: $F_{(1,27)}$ = 9.57, p=0.005; interaction $F_{(1,27)}$ = 0.38, p=0.5), but this difference

disappeared when the shock was turned off during the extinction trials (genotype: $F_{(1,27.9)}$ = 0.26, p=0.6; trials: $F_{(1,27.4)}$ = 0.51. p=0.5; interaction $F_{1,27.4}$ = 0.35. p=0.6) (EX1, EX2). When exploring the novel environment with shock off (nPRE), the mutant mice walked ~25% less than the controls ($F_{(1,23.3)}$ = 4.37, p=0.048), but the difference was not observed once training in the new environment began (genotype: $F_{(1,29.1)}$ = 0.01, p=0.9; trial $F_{(2,31.2)}$ = 10.40, p=0.0003; interaction $F_{(2,31.2)}$ = 0.37, p=0.7) (nT1-nT3). These analyses show that, in some tasks, locomotion differed by as much as ~25% between mutant and control mice, but it did not appear to vary in a systematic manner. Because the physical environment is essentially the same across the place avoidance protocol, these differences likely reflect internal cognitive variables rather than genotypic differences in particular sensory or motor abilities.

## Phenotype of the *Wntless Gfap-Cre* conditional mutant in an active avoidance task: learning and memory

On day 2, all of the mice learned to avoid the shock zone, but active place avoidance learning was compromised in mutant mice (*Figure 6D*). During pretraining without shock (PRE), mutant mice entered the future location of the shock zone less often (*Figure 6D*, left panel; $F_{(1,26)}$ = 17.76, p=0.0003), presumably because they walked less (*Figure 6C*), but the genotypes did not differ in the latency to enter the future shock zone (*Figure 6D*, center panel; $F_{(1,26)}$ = 0.64, p=0.4) or the maximum time spent outside of the future shock zone (*Figure 6D*, right panel; $F_{(1,25.9)}$ = 2.44, p=0.1). Learning the place avoidance response (iT1-iT3) was reduced in mutant mice, as measured by the number of entrances (*Figure 6D*, left panel; genotype: $F_{(1,26)}$ = 7.90, p=0.009; trial: $F_{(2,29.9)}$ = 25.43, iT1 >iT2=iT3; p=$10^{-7}$; interaction: $F_{(2,29.9)}$ = 1.23, p=0.3), the time to first entrance (*Figure 6D*, center panel; genotype: $F_{(1,39.8)}$ = 1.96, p=0.17; trial: $F_{(2,38.1)}$ = 7.90, p=0.001, iT1 <iT3; interaction: $F_{(2,38.1)}$ = 1.61, p=0.2), and the maximum avoidance time (*Figure 6D*, right panel; genotype: $F_{(1,27)}$ = 4.04, p=0.054; trial: $F_{(2,33.3)}$ = 3.68, p=0.04, iT1 <iT3; interaction: $F_{(2,33.3)}$ = 0.39, p=0.7), although the differences between genotypes for the latter two measures did not reach statistical significance (i.e. p<0.05).

The 24 hr memory retention test (RET) showed that all the mice had learned, but the mutant mice exhibited a modestly poorer performance than the controls (*Figure 6D*), with a greater number of entrances ($F_{(1,27)}$ = 3.66, p=0.07), a shorter time to first enter the shock zone ($F_{(1,27)}$ = 3.84, p=0.06), and a shorter maximum avoidance time ($F_{(1,27)}$ = 4.66, p=0.04).

Cognitive flexibility was assessed by conflict trials in which the shock zone was relocated 180° (CO1 and CO2) (*Figure 6E*). By comparing retention performance before and after the change, it is evident that the relocation disrupted place avoidance, confirming that both genotypes had learned a place response. Learning the new location of the shock zone was not measurably poorer for the mutant mice as assessed by the number of entrances into the relocated shock zone (*Figure 6E*, left panel; genotype: $F_{(1,27)}$ = 2.26, p=0.14; trial: $F_{(1,27)}$ = 16.78, p=0.0003; interaction: $F_{(1,27)}$ = 0.58, p=0.5). For both genotypes, the time to first enter the relocated shock zone increased only modestly across the two trials (*Figure 6E*, central panel; genotype: $F_{(1,27)}$ = 0.38, p=0.5; trial: $F_{(1,27)}$ = 3.69, p=0.07; interaction: $F_{(1,27)}$ = 0.16, p=0.7). Whereas the control mice increased the maximum avoidance time across the two conflict trials, the mutant mice did not (*Figure 6E*, right panel; genotype: $F_{(1,27)}$ = 3.01, p=0.09; trial: $F_{(1,28.5)}$ = 4.81, p=0.04; interaction: $F_{(1,27)}$ = 4.22, p=0.049).

When the shock was turned off (EX1 and EX2), both mutant and control mice began to extinguish the place response as assessed by the number of entrances into the previous shock zone (*Figure 6F*, left panel; genotype: $F_{(1,28)}$ = 0.13, p=0.7; trial: $F_{(1,27.8)}$ = 0.17, p=0.7; interaction: $F_{(1,27.8)}$ = 0.69, p=0.4), the time to first enter the previous shock zone (*Figure 6F*, central panel; genotype: $F_{(1,27.1)}$ = 1.84, p=0.2; trial: $F_{(1,27.5)}$ = 0.62, p=0.4; interaction: $F_{(1,27.5)}$ = 0.022, p=0.9), and the maximum avoidance time (*Figure 6F*, right panel; genotype: $F_{(1,28.2)}$ = 0.0091, p=0.9; trial: $F_{(1,29.1)}$ = 1.25, p=0.3; interaction: $F_{(1,29.1)}$ = 0.087, p=0.8).

Two days later, the mice were tested in a new arena in a different visual environment. During pretraining (nPRE), the two genotypes showed similar performances (*Figure 6G*; Entrances: $F_{(1,27)}$ = 2.17, p=0.2; Time to first entrance: $F_{(1,27)}$ = 2.52, p=0.1; maximum avoidance time $F_{(1,27)}$ = 0.76, p=0.4). Activating the shock zone in the new environment revealed that both mutant and control mice learned to avoid the shock zone, as determined by a reduced number of entrances (*Figure 6G*, left panel; genotype: $F_{(1,27.2)}$ = 0.71, p=0.4; trial: $F_{(2,27.9)}$ = 3.61, p=0.04, nT1 >nT3; interaction: $F_{(2,27.9)}$ = 0.98, p=0.4), a progressive increase in the time to first enter the shock zone (*Figure 6G*,

central panel; genotype: F(1,51.3) = 0.34, p=0.6; trial: F(2,44.5) = 8.12, p=0.001, nT1 <nT3; interaction: F(2,44.5) = 1.21, p=0.3), and an increase in the maximum avoidance time (*Figure 6G*, right panel; genotype: F(1,27.8) = 2.23, p=0.15; trial: F(2,36.4) = 8.02, p=0.001, nT1 <nT2=nT3; interaction: F(2,36.4) = 0.14, p=0.9). In the latter two tests, the mutant mice exhibit a modestly reduced performance compared to controls, but the differences did not achieve statistical significance. In sum, the active avoidance tests show a consistent and modest reduction in conditioned spatial learning and memory in *Wls^{fl/-};Gfap-Cre* compared to *Wls^{fl/+};Gfap-Cre* mice.

Two days after the end of place avoidance training, the mice were tested in a T-maze alternation task to assess non-spatial cognitive ability (*Figure 7*). Each mouse was trained to escape to one arm during 10 trials, and then, after a 15 min rest in the home cage, trained to escape to the opposite arm. This left/right alternation was repeated in blocks of 10 trials until 40 trials were completed. In the first trial, the performances of the two genotypes were indistinguishable. In the second and third trials, the mutant mice performed marginally better than the controls, a difference that was not statistically significant. However, in the fourth trial, the performance of the controls was significantly better than that of the mutants. (*Figure 7*; genotype: F(1,34.8) = 0.005, p=0.9; session: F(3,49.3) = 7.39, p=0.0003; genotype x trial interaction: F(3,49.3) = 3.98, p=0.01; mutant (session 4) >control (session 4).) A parsimonious explanation of these results is that the decrement in the performance of the mutant mice on the fourth trial arises from the increase in cognitive demand associated with greater numbers of alternations.

## Discussion

The experiments reported here introduce a new mouse model, *Wls^{fl/-};Gfap-Cre*, that we have used to test the role of DGCs in spatial learning and memory. The neuroanatomic defects in adult *Wls^{fl/-};Gfap-Cre* mice consist of a complete loss of the corpus callosum and a ~ 90% loss of DGCs, with most of the remaining DGCs localized to the ventral DG. The reduction in mature DGCs is associated with a reduction in Wnt signaling in the cortical hem and a reduction in DGC progenitor proliferation. *Wls^{fl/-};Gfap-Cre* hippocampi shows minimal alterations in cell-type-specific gene expression, as determined by bulk and snRNAseq, implying that the developmental reduction in DGCs has little or no effect on the developmental trajectories of other hippocampal cells, including CA3 pyramidal cells, the recipients of synaptic input from DGC-derived mossy fibers. Quantification of CA3 pyramidal cell morphology and synaptic density/structure by Golgi staining showed only a modest reduction in the density of complex synapses in *Wls^{fl/-};Gfap-Cre* mice. Behavioral testing revealed that *Wls^{fl/-};Gfap-Cre* mice have a modest performance decrement in complex spatial learning and memory tasks, but *Wls^{fl/-};Gfap-Cre* mice also exhibit impaired performance in one simpler spatial task – finding a visible platform in the MWM training sessions. Whether or to what extent the loss of the corpus callosum or the modest changes in the abundances of non-DGC neurons in *Wls^{fl/-};Gfap-Cre* mice might contribute to the observed learning and memory defects is currently an open question. As discussed more fully below, the dorsal hippocampus has been implicated in spatial, as opposed to olfactory, information processing, and, therefore, the nearly complete absence of DGCs in the dorsal hippocampus in *Wls^{fl/-};Gfap-Cre* mice makes this model especially well-suited for studying the contributions made by DGCs to spatial tasks.

### Canonical Wnt signaling and the development of the dentate gyrus

The *Wls^{fl/-};Gfap-Cre* DG phenotype represents one point along a continuum of hippocampal defects resulting from reduced canonical Wnt signaling in the cortical hem, which is the source of both Wnt ligands and DGC progenitors. Eliminating *Lrp6* leads to an absence of ~50% of DGCs, eliminating *Lef1* leads to an absence of nearly all DGCs, and eliminating *Wnt3a* (which is specifically expressed in the cortical hem) or replacing *Lef1* with a gene coding for a dominant negative derivative that inhibits beta-catenin-dependent gene activation by other LEF/TCF family members leads to a complete absence of the entire hippocampus (*Galceran et al., 2000*; *Lee et al., 2000*; *Zhou et al., 2004*). The high concentration of LEF1 in the cortical hem is consistent with a role for canonical Wnt signaling in driving proliferation among DGC progenitors. In addition to this role, canonical Wnt signaling may also have an instructive role in cell specification in the developing cortex, with a gradient of Wnt signaling generating a series of distinct neuronal fates (*Machon et al., 2007*).

## Molecular and regional diversity among dentate granule cells

Our snRNAseq analysis has revealed DGC transcriptome diversity along at least two distinct dimensions, one of which corresponds to the dorsal/ventral axis of the hippocampus. Transcriptome diversity along the dorsal/ventral axis could be related to functional diversity along this axis, as revealed by electrophysiological studies (*Papatheodoropoulos, 2015*; *Kouvaros and Papatheodoropoulos, 2017*), by physiological effects on target regions beyond the hippocampus (*Sosa et al., 2020*), and by lesion studies that have implicated the dorsal hippocampus in spatial processing and the ventral hippocampus in anxiety and olfactory learning (*Kesner et al., 2011*; *Strange et al., 2014*; *Hauser et al., 2020*). Although visual inspection of Allen Brain Atlas ISH patterns does not reveal large-scale anatomic correlates for DGC transcriptome diversity other than the dorsal/ventral pattern, local DGC heterogeneity is suggested by the functional heterogeneity of hippocampal CA3 pyramidal cells along the transverse and radial dimensions, which includes differences in the distribution of DGC inputs (*Cembrowski et al., 2016*). Additional sources of local DGC heterogeneity are the presence of adult-born DGCs at various stages of maturation (*Wang et al., 2000*; *Chatzi et al., 2016*) and changes in DGC gene expression that reflect recent changes in electrical activity (*Ramirez-Amaya et al., 2013*).

## Resilience of the hippocampus to developmental loss of dentate granule cells

A striking feature of the *Wls$^{fl/-}$;Gfap-Cre* hippocampus is the minimal effect of a congenital reduction in DGC number on the transcriptomes of other hippocampal cell types. The modest changes in the number and type of CA3 pyramidal cell synapses suggest that the near absence of mossy fiber input is, at least partially, compensated by an increase in inputs from other neurons, potentially from entorhinal cortex. These features are reminiscent of other examples of robust developmental trajectories in the context of early perturbations in CNS structure or activity. For example, in the classic monocular deprivation experiments of Hubel and Wiesel, inputs to primary visual cortex from the normal eye (via the lateral geniculate nucleus) expanded to compensate for the reduced input from the sutured eye when the perturbation occurs in early postnatal life (*LeVay et al., 1980*). Histologic analyses of primary visual cortex showed that the cellularity and total synapse density were largely unchanged in those regions of visual cortex that normally would have received equal binocular inputs (*Shatz and Stryker, 1978*; *Silver and Stryker, 1999*). However, in other contexts, developmental perturbations lead to uncompensated changes, such as the increased cell death among dorsal root ganglion neurons that follow the surgical removal of target tissues, such as a limb, during embryonic life (*Hamburger, 1992*).

The developmental hypoplasia of DGCs in *Wls$^{fl/-}$;Gfap-Cre* mice offers an alternative to existing models of DGC ablation in the mature CNS. In contrast to early developmental perturbations, the loss of mature CNS neurons is generally accompanied by reactive gliosis and by perturbations in the structure, function, and/or viability of synaptically-linked neurons (*Burda and Sofroniew, 2014*; *Pfeiffer et al., 2020*). Such secondary effects can complicate the interpretation of any resulting physiological or behavioral changes. The modest degree of secondary cellular changes in the *Wls$^{fl/-}$; Gfap-Cre* hippocampus suggests that this model will be useful for future electrophysiological and optical interrogations of hippocampal circuit responses to a reduction in DGC inputs.

## Role of the dentate gyrus in spatial learning and memory

Impaired performance in the cognitive tasks employed in this study is consistent with a model in which DGCs enhance performance in the context of complex spatial tasks. However, as noted above, the impaired performance of *Wls$^{fl/-}$;Gfap-Cre* mice in a simpler spatial task – finding a visible platform in the MWM pretraining sessions – suggests that the DGC loss results in broader cognitive deficits. It would be interesting to determine if increasing the contextual richness of the cognitive tasks would reveal an even greater effect of DGC input. For example, a recently developed behavioral paradigm in which a mouse runs repeatedly around a topologically closed maze demonstrates that calcium responses in a subset of hippocampal CA1 pyramidal cells were specific to both spatial location and lap number (*Sun et al., 2020*). This paradigm could be used to determine the extent to which the precision of lap counting depends on DGC function.

In summary, the *Wls*$^{fl/-}$*;Gfap-Cre* mice described here provide the community with an approach to selective DGC reduction that (i) is distinct from the less specific colchicine ablation and neonatal irradiation approaches that have been used over the past several decades (*Xavier and Costa, 2009*) and (ii) is well characterized at the levels of single-cell gene expression and CA3 pyramidal cell morphology.

## Materials and methods

### Mice

Mice used for this study are as follows: *Wls*$^{fl}$ (*Carpenter et al., 2010*; JAX stock no. 012888) and *Tg (Gfap-Cre) 25Mes/J* (*Zhuo et al., 2001*; JAX stock no. 004600).

Genotyping primers used for this study are as follows: *Gfap-Cre*, AR1382 (oIMR1900), 5′-ACTCC TTCATAAAGCCCT-3′ and AR1383 (oIMR1901), 5′-ATCACTCGTTGCATCGACCG-3′; *Wls*$^{fl}$ allele, P2, 5′-AGGCTTCGAACGTAACTGACC-3′ and P4, 5′-CTCAGAACTCCCTTCTTGAAGC-3′; *Wls*$^{KO}$ allele, P1, 5′-CTTCCCTGCTTCTTTAAGCGTC-3′ and P4, 5′-CTCAGAACTCCCTTCTTGAAGC-3′.

### Diffusion tensor magnetic resonance imaging

Specimen preparation. For ex vivo MRI, brains of age-matched adult *Wls*$^{fl/-}$*;Gfap-Cre* and *Wls*$^{fl/+}$*; Gfap-Cre* mice (three mice per genotype) were perfusion fixed in 4% paraformaldehyde (PFA) followed by overnight immersion in 4% PFA. Prior to imaging, the brains were transferred to phosphate buffered saline (PBS) with 2 mM gadopentetate dimeglumine (Gd-DTPA, Berlex Imaging, Wayne, NJ, USA) for 72 hr, and then placed in 15 mm diameter glass tubes that were filled with perfluoropolyether (Fomblin, Solvey Solexis, Thorofare, NJ, USA) to prevent dehydration.

Imaging. MRI of the mouse brains was performed on a vertical-bore 11.7 T scanner (Bruker Biospin, Billerica, MA, USA) equipped with a Micro2.5 gradient system. A 15 mm diameter birdcage coil was used for signal transmission and reception. The temperature of the brains was maintained at 37℃ during imaging via thermostatically-controlled airflow integrated with the scanner. Diffusion MRI data were acquired using a three-dimensional diffusion-weighted gradient-and-spin-echo (DW-GRASE) sequence with twin navigator echoes (*Aggarwal et al., 2010*), using the following imaging parameters: (diffusion gradient duration)/(separation) = 3.5/15 ms, echo time (TE) = 28 ms, repetition time (TR) = 800 ms, and two signal averages. For each brain, diffusion-weighted images along 30 independent directions (b-value = 2000 s/mm$^2$) and two non-diffusion-weighted images were acquired with a spatial resolution of 70 x 70 x 70 µm. The total imaging time for each brain was ~13 hr.

Image analysis. Images were reconstructed using MATLAB (Mathworks Inc, Natick, MA, USA). The k-space data were zero-filled to twice the matrix size prior to Fourier transformation. Diffusion tensors were calculated using the log-linear fitting function in DtiStudio (www.mristudio.org). For analysis, all images were aligned to one wild-type mouse brain chosen as the anatomical reference, using intensity-based linear rigid registration based on the non-diffusion-weighted images, followed by non-linear registration using large deformation diffeomorphic metric mapping (*Miller et al., 2002*). The derived transformations were then used to spatially normalize and reorient the diffusion tensors using the methods described in *Alexander et al., 2001*. From the averaged diffusion tensors of the control and mutant mouse brains, parametric fractional anisotropy (FA), primary eigenvector, and direction-encoded color maps were calculated for each group (*Aggarwal et al., 2015*).

### Antibodies

Antibodies used in this study were as follows: rabbit anti-LEF1 mAb (2230S, clone C12A5; Cell Signaling), chicken anti-GFP (ab13970; Abcam), rabbit anti-Calbindin D-28k (CB-38a; Swant), rabbit anti-calretinin (7697; Swant), goat anti-Prox-1 (AF2727; R&D Systems), and goat anti-Reelin (AF3820; R&D Systems). Alexa Fluor–labeled secondary antibodies were from Invitrogen.

### Tissue processing and immunohistochemistry

Tissues were prepared and processed for immunohistochemical analysis as described previously (*Wang et al., 2012*; *Zhou et al., 2014*). Briefly, embryonic brains were immersion fixed overnight at 4℃ in 1% paraformaldehyde (PFA), followed by 100% MeOH dehydration overnight at 4℃. Adult

mice were perfused transcardially with 4% PFA in PBS, the brains were dissected out of the skull and post-fixed in 4% PFA in PBS for several hours to overnight at 4°C followed by 100% MeOH dehydration overnight at 4°C. All tissues were rehydrated the following day in 1 × PBS at 4°C for at least 3 hr before embedding in 3% agarose. 200 μm (adult) and 100 μm (embryonic) brain sections were cut using a vibratome (Leica).

Tissue sections were incubated overnight with primary antibodies (1:500) in PBSTC (1 × PBS + 0.5% Triton X-100 + 0.1 mM CaCl$_2$) plus 10% normal goat or normal donkey serum. Tissues were washed three times with PBSTC over 6–8 hr and then incubated overnight with secondary antibodies (1:500) diluted in 1x PBSTC + 10% normal goat or normal donkey serum. Tissues were then washed at least three times with PBSTC over 6 hr, flat mounted using Fluoromount G (EM Sciences 17984–25), and imaged using a Zeiss LSM700 confocal microscope using Zen Black 2012 software.

## Bulk RNAseq

Two biological replicates were sequenced per genotype. For each sample, RNA from both hippocampi from a single male mouse was extracted with TRIzol (Invitrogen 15596026) followed by purification with the RNeasy Mini kit (QIAGEN 74104). Libraries were constructed with the NEBNExt Ultra II directional library prep kit (NEB E7760L) and sequenced on an Illumina Hiseq2500. Read alignment to the mm10 reference mouse genome sequence was performed with the RSEM-1.3.0 program (*Li and Dewey, 2011*) using the Bowtie2-2.2.9 aligner (*Langmead and Salzberg, 2012*). Differential gene expression analysis was performed with EBseq 1.24.0 (*Leng et al., 2013*).

## snRNAseq

Two biological replicates were sequenced per genotype. For each sample, both hippocampi of a single male mouse were rapidly dissected in ice-cold homogenization buffer (0.25 M sucrose, 25 mM KCl, 5 mM MgCl$_2$, 20 mM Tricine-KOH, pH=7.8). The tissue was minced with a razor blade and Dounce homogenized using a loose-fitting pestle in 5 ml of homogenization buffer supplemented with 1 mM DTT, 0.15 mM spermine, 0.5 mM spermidine, EDTA-free protease inhibitor (Roche 11 836 170 001), and 60 U/mL RNasin Plus RNase Inhibitor (Promega N2611). A 5% IGEPAL-630 solution was added to bring the homogenate to 0.3% IGEPAL-630, and the homogenate was further homogenized with five strokes of a tight-fitting pestle. The sample was filtered through a 50 μm filter (CellTrix, Sysmex, 04-004-2327), underlayed with solutions of 30 and 40% iodixanol in homogenization buffer (Sigma D1556), and centrifuged at 10,000 x g for 18 min in a swinging bucket centrifuge at 4°C. Nuclei were collected at the 30–40% interface, diluted with two volumes of homogenization buffer and concentrated by centrifugation for 10 min at 0.5 x g at 4°C. snRNAseq sequencing libraries were constructed using the 10X Genomics Chromium single cell 3' v3 kit. Libraries were sequenced on an Illumina NovaSeq 6000.

## Analysis of snRNAseq data

Reads were aligned to a mm10 pre-mRNA index using Cellranger version 3.1.0. Libraries were merged using Cellranger and data was analyzed using both the Monocle 3 (*Qiu et al., 2017*) and Seurat 3.1 (*Butler et al., 2018*) R packages, with similar results. Using Monocle 3, expression data was log-normalized (with a pseudo-count of 1) and the lower dimensional space was calculated using principal component analysis (PCA). Batch effects were corrected using the mutual nearest neighbor algorithm as described (*Haghverdi et al., 2018*). The Uniform Manifold Approximation and Projection (UMAP) algorithm was used for two-dimensional reduction of the data (*Becht et al., 2019*). Cells were clustered using the Monocle 3 cluster_cells method, based on Louvain/Leiden community detection with default settings, with UMAP reduction as input. To identify transcript expression modules within the cluster of DGCs, we used the Monocle 3 graph_test algorithm (monocle3::graph_test) that implements Moran's I statistics to identify pattern of expression in a two-dimensional reduced expression data. To test for differences in transcripts, the Monocle 3 implementation of regression analysis (monocle3::fit_models) was used both globally and separately on each identified cell type. For analysis with the Seurat R package, the expression data was normalized using a regularized negative binomial regression as described in *Hafemeister and Satija, 2019*. Data exploration, analysis, and plotting were performed using RStudio (*R Studio, 2016*), the tidyverse collection of R packages (*Wickham, 2017*), and ggplot2 (*Wickham, 2009*).

## Neuron reconstruction and characterization of morphological features of CA3 pyramidal neurons

To investigate the morphology of CA3 pyramidal neurons, two $Wls^{fl/+}$;$Gfap$-$Cre$ brains and two $Wls^{fl/-}$;$Gfap$-$Cre$ brains where treated with the Rapid Golgi kit (FD NeuroTechnologies) and cut at a thickness of 120 µm. The tissue was analyzed using Neurolucida on a Zeiss Axio Imager1 with an automated stage and a 100x oil objective (Numerical Aperture 1.2). The hippocampus was delineated by visual inspection of the cytoarchitecture (*van Strien et al., 2009*). Cells were chosen for reconstruction in area CA3b (the sharpest curve of the hippocampus). A neuron was classified for reconstruction if it possessed CA3 pyramidal cell morphology, if it had one apical dendrite and at least one basal dendrite available for tracing, and if it was located in area CA3b. Neurons were excluded if their position was in close proximity to an astrocyte or if the density of Golgi-impregnated cells in its immediate proximity was too high. Spines were either classified as thorny excrescences (complex) or as 'other' (simple) spines based on spine shapes as described in *Sorra and Harris, 2000*. Using criteria from *Gonzales et al., 2001*, a cluster of thorny excrescences along the dendrite was classified as a complex spine, rather than classifying each individual protrusion. Once a complex spine was identified, the maximum extent of the part of the dendrite that is covered by the spine was measured using the 'spherical spine tool' in Neurolucida (MBF BioScience). The diameter of the spine-tool was adjusted until it covered the maximum length that the complex spine extended along the dendrite. This variable is referred to as spine length. Nine $Wls^{fl/+}$;$Gfap$-$Cre$ and nine $Wls^{fl/-}$;$Gfap$-$Cre$ CA3 pyramidal neuron reconstructions were analyzed by extracting data from the Neuron Summary and Spine Detail functions in Neurolucida Explorer.

## Data visualization and statistical analysis

The morphological data from the neuron reconstructions were extracted from a Branched Structure Analysis in Neurolucida Explorer (MBF Bioscience). The variables extracted and used for further analysis were from the 'Neuron Summary' and 'Spine Detail' analysis option of the Branched Structure Analysis. More specifically, the length of dendrites for both apical and basal dendrites, the different spine types (simple and complex), and the number of spines per dendrite type was extracted from the 'Neuron Summary' analysis, while the 'Spine diameter' of all individual spines was extracted from the 'Spine detail' analysis. This 'Spine diameter' refers to the spine length described above. Spine density was calculated for individual dendrites by dividing the total number of spines on a dendrite by the length of that dendrite. Spine coverage on the total dendritic length represents the summed spine length divided by the summed length of all the dendrites. The final dataset was exported to MATLAB for further analysis, which included the creation of boxplots using the boxplot function and a Wilcoxon Rank Sum non-parametric hypothesis test using the ranksum function (MATLAB, 2019). The Wilcoxon Rank Sum was chosen based on inspection of plots of the data fitted to a normal distribution (normplot function, MATLAB, 2019) that indicated that every dataset had some, if not all variables violating the assumption of normally distributed data needed to conduct a student t-test.

## Stereology

To compare the presence of thorny excrescences (complex spines) along the dorsoventral extent of hippocampal area CA3 in $Wls^{fl/+}$;$Gfap$-$Cre$ vs. $Wls^{fl/-}$;$Gfap$-$Cre$ mice, a stereological investigation was conducted. Golgi-stained sagittal sections ($Wls^{fl/+}$;$Gfap$-$Cre$, N = 25; $Wls^{fl/-}$;$Gfap$-$Cre$, N = 26) were analyzed from one hemisphere for each genotype. The Stereo Investigator (MBF Bioscience) was used on a Axio Imager1 (Zeiss) with an automated stage and a 100x oil objective (Numerical Aperture 1.2). The optical fractionator method was used to conduct a designed-based two-stage systematic sampling (*Gundersen, 1986*; *Gundersen et al., 1988*; *West and Gundersen, 1990*; *West et al., 1991*).

The hippocampus was traced in all sections where it was present. The relevant layers were delineated by visual inspection using darkfield microscopy to visualize the pyramidal cell layer. The stratum lucidum was standardized and delineated across sections by taking the width of the pyramidal cell layer and adding a similarly shaped layer with the same width directly superficial to the pyramidal cell layer. All delineations were confirmed by an experienced researcher (M.W.) prior to starting the stereological counting. In view of the known distribution of the mossy fiber projection in stratum lucidum as well as in stratum pyramidale, dendritic branches were sampled in both layers. Based on

pilot runs conducted in two hemispheres and tests of oversampling and subsampling in one section, an oversampling approach was selected. *Supplementary file 1* shows the sampling parameters chosen for the final stereological run.

Thorny excrescences (complex spines) were classified based on *Gonzales et al., 2001*. Five complex spine subtypes were defined as follows. A 'basic complex spine' was defined as being >3 μm along the dendrite and >2.5 μm in height (*Gonzales et al., 2001*; *Tsamis et al., 2010*) and located on the first or second order branch transitions, which is typical of thorny excrescences (*Amaral, 1978*). A 'big/prototypical complex spine' was defined as being in the transition between the first and second order branches and >8 μm in length or consisting of more than two defined complex spines in such close proximity that they were difficult to distinguish. A 'long complex spine' was defined as a complex spine that extended >3 μm along the dendrite and <2 μm in height. A 'tall complex spine' was defined as a complex spine that extended <3 μm along the dendrite and >2 μm in height. A 'thin complex spine' was defined as a complex spine in which individual branches and spine heads could be resolved.

All stereological runs achieved an acceptable Gundersen coefficient of error (CE) (*Glaser and Wilson, 1998*), indicating that the sampling was representative. The estimated area, volume, and mean thickness of sections was approximately the same between the *Wls^{fl/+};Gfap-Cre* and *Wls^{fl/-};Gfap-Cre* samples. *Supplementary file 2* shows that the total area is slightly larger in the *Wls^{fl/-};Gfap-Cre* sample and the estimated volume is slightly larger in the *Wls^{fl/+};Gfap-Cre* sample. This apparent inconsistency is explained by the four missing sections in the *Wls^{fl/+};Gfap-Cre* sample, since the total area represents the actual delineated area whereas the estimated volume takes into account the missing sections.

## Statistical analysis

All the counted spines from the stereological run from both animals was organized in a datasheet to test if there was any significant differences in the mean number of counted spines between the *Wls^{fl/+};Gfap-Cre* hemisphere and *Wls^{fl/-};Gfap-Cre* hemisphere. Since a test of normality showed that the data was non-normal, we chose the Mann-Whitney-Wilcoxon test.

## Behavioral testing: baseline neurological testing

At Johns Hopkins University, three standard neurological tests were administered.

- Open field. Locomotor activity was assessed over 30 min in a 40 × 40 cm activity chamber with infrared beams (San Diego Instruments). Horizontal activity, as well as time spent in the center or periphery of the chamber, was automatically recorded.
- Rotarod. Mice were placed on the rotarod with a starting speed of 4 rpm and an acceleration of 6 rpm/min. The time at which each mouse dropped from the rotating rotarod was recorded. Over three days, each mouse was given three trials per day with a 2 min inter-trial interval.
- Elevated plus maze. Anxiety-related behavior was evaluated using the elevated plus maze test. Mice were placed in the center of a 54 cm high maze consisting of two open and two closed 66 cm long arms for 5 min. Distance traveled and time spent in the open and closed arms was automatically recorded using Topscan tracking software (Cleversys).

At New York University, seven standard neurological tests were administered to each mouse prior to the active place avoidance test battery.

- Vertical screen test. The mouse was placed on a square, horizontally-oriented flat cage top with 1 cm spacing between the bars. The cage top was slowly rotated 90° to a vertical position and whether the mouse showed a climbing response within 30 s was recorded.
- Negative geotaxis test. The same cage top was used as in the vertical screen test. The cage top was placed at a 45° angle on a desk. The mouse was placed on the cage top, facing downwards. The latency to turn and orient with the head facing upwards was recorded. Latency times were averaged over three trials.
- Parallel bars. A pair of bars (1 m long, 3 mm diameter) were placed at a height of 50 cm and parallel to each other, with 3 cm spacing between them. The mouse was placed in between the two bars, perpendicularly, and the time to grasp the bars and turn 90° was recorded, up to 30 s. Latency times were averaged over two trials.
- Visual placing response. The mouse was suspended by the tail and moved downward near to the edge of a table. The experimenter noted whether the mouse reached to grab onto the

table edge. The reflex was tested three times with each mouse receiving a score between 0 and 3.

- Body suspension test. The mouse was held by the tail, so it could grasp and hang from a 4 mm diameter bronze bar (20 cm long and suspended 30 cm above the bench surface). The time to drop from the bar was recorded, up to 30 s. If the mouse climbed up onto the bar, the time was counted as 30 s. The test was repeated three times.
- Spontaneous grooming. A mouse was placed in a novel cage and its behavior was recorded for 5 min from the side with a cellphone video camera. The total time the mouse was observed to be grooming was recorded. Grooming behavior was classified into four categories: 1, washing front paws, snout, and head; 2, licking back and front body fur; 3, scratching head with hind paw and licking hind paw; and 4, washing tail.
- Elicited grooming. Five minutes after the spontaneous grooming test, each mouse was sprayed twice with water. Behavior was recorded for 5 min with a cellphone video camera and scored as in the spontaneous grooming test.

## Behavioral testing: spatial cognition and memory

### Y-maze spatial recognition and memory

The Y-maze consists of three 38 cm-long arms (San Diego Instruments). During the training phase, one arm of the Y-maze was blocked. The mouse was placed at the end of one of the two open arms and allowed to explore for 5 min. After a 30-min inter-trial interval, the test phase began: the blockade was removed, and the mouse was allowed to explore all three arms of the maze for 5 min. Distance traveled and time spent in each arm was automatically recorded using Topscan tracking software (Cleversys). Data from the first 2 min of the test phase were used to evaluate percent time spent in the novel arm.

### Barnes maze

The Barnes maze test was based on the protocol used by *Rahn et al., 2012*. Briefly, a brightly lit (1100 lux) Barnes maze with 40 evenly spaced holes and an escape box placed under one of the holes was used (Maze Engineers). During training, each mouse was placed in the center of the maze and allowed to explore the maze for 3 min per trial. During the trial, the number of head dips and the latency to find and then enter the escape box were recorded. Mice were given four trials per day for four days. 48 hr following training, mice were given a probe trial, with the number of head dips and latency to find and then enter the escape box recorded.

### Trace fear conditioning

Trace fear conditioning was conducted as previously described in *Terrillion et al., 2017*. Briefly, over three consecutive days, trace fear conditioning consisted of a habituation day, a training day, and a test day. On the habituation day, the mouse was exposed to the shock box (Coulbourn) for 10 min. On the training day, the mouse was placed in the shock box and given a 2-min habituation, after which a 20 s white noise tone (80 db, 2000 Hz) was delivered. Twenty seconds following the termination of the tone, a scrambled 2-s 0.5 mA shock was delivered. The tone-shock pairing was repeated three additional times. On the test day, the mouse was placed in the shock box for 3 min to measure freezing in response to context. The mouse was then placed in a separate context and freezing in response to the 20 s white noise tone was measured. Freezing behavior was automatically scored using Freezescan software (Cleversys).

### Morris water maze

The Morris water maze test (MWM) was based on the protocol previously described in *Pletnikov et al., 2008*. The maze consisted of a circular stainless-steel tank 4 m in diameter filled with room temperature water made opaque with white tempera paint for the training, probe, and reversal trials. During the pretraining phase, a 10 cm platform was placed 1 cm below clear water in the center of a 2-m diameter stainless-steel cylinder in the MWM. For each mouse, the test protocol was as follows. The mouse was placed inside the perimeter of the cylinder and given 60 s to escape onto the visible platform throughout six trials in one day. During the hidden platform training phase, a 10 cm platform was placed in a fixed location in the maze, with the top of the platform hidden beneath 1 cm of water. The mouse was placed in the maze around the perimeter in one of four start

positions in a semi-random fashion throughout six trials each day for three consecutive days. The mouse was allowed to search for the platform for 60 s, and, after finding the platform, to remain there for 15 s. Escape latency was recorded for each trial. 24 hr following the hidden platform training phase, the mouse was tested in one 60 s probe trial. Latency to cross into the platform area, number of crossings in the platform area, and time spent in the platform quadrant were measured using Anymaze tracking software (Stoelting). During the reversal phase, the hidden platform was placed in the MWM in a different quadrant from that used during the training phase. As in the training phase, the mouse was placed in the maze around the perimeter in one of four start positions in a semi-random fashion throughout six trials each day for two consecutive days. The mouse was allowed to search for the platform for 60 s and, after finding the platform, to remain there for 15 s. Escape latency was recorded for each trial. Path efficiency to the target was by calculating (i) the length of a straight line from the location where the mouse was placed in the tub to the platform, (ii) the distance that the mouse swam from its starting location to the first time it arrived at the platform ('arrival' being defined as the center of the mouse's body crossing an 8 cm diameter circle centered on the platform), and (iii) the fraction consisting of the first value divided by the second value, giving a number between 0 and 1.

## Place avoidance training

The place avoidance apparatus and testing protocol are based on those described in *Cimadevilla et al., 2001* and *Kubík and Fenton, 2005*.

### Apparatus

 The apparatus was placed in the vivarium 2 m from the rack that housed the mouse cages. A 40 x 40 cm floor made of parallel stainless-steel rods was used. The grid floor was elevated 70 cm on a motorized turntable that could rotate at 1 rpm. A clear plastic cylindrical wall inscribed a circular space on the floor to contain the mouse. While in the test arena, the mouse could see multiple landmarks in the room including shelves, desks, and a pair of poles supporting a curtain rod on which white plastic curtains hung to visually separate the apparatus from the experimenter and rack of cages. An overhead digital video camera was connected to a computer running video- tracking software (Tracker; Bio-Signal Group) to determine the position of the mouse at 33 millisecond intervals. The software could trigger a mild foot shock that was scrambled across the five electric poles of the grid floor. Two test arenas were used. They differed in their location in the room and one arena had plastic ties on some of the bars on the grid floor to distinguish it. Mice were transported between the home cage and test arena in a small plastic cup.

### Experimental design and protocol

Training took place during the light phase of a 12:12 (light:dark) cycle, with lights on at 7 AM. There were five behavioral phases, each designed to evaluate one aspect of spatial behavior across multiple trials. Each trial was 10 min in duration. No physical changes were made to the test arena during the first four phases except for the presence or absence of a foot shock.

- Open field. The arena was stationary. On day 1, each mouse was allowed to explore the arena for 10 min on two trials separated by 1 hr. The mouse was returned to its home cage between trials. The distance traveled in the equal-area circle and annulus of the arena was measured.
- Initial Training. The arena was rotating at 1 rpm. On day 2, pretraining began and each mouse was allowed to explore the arena for 10 min with no shock, as in the open field test conditions, except the arena was rotating. After a 1 hr rest in the home cage, active place avoidance training began. The mouse was placed in the apparatus for three trials and returned to its home cage for the 1 hr separation between trials. The environment was identical to the pretraining condition, except that the mouse received a mild 500 ms, 600 Hz, 0.2 mA constant current foot shock if it entered a 60° sector that was designated the shock zone. The shock was repeated every 1.5 s until the mouse left the shock zone. The track of the mouse was stored and automatically analyzed offline with TrackAnalysis software (Bio-Signal Group). The total distance walked on the arena surface was computed to evaluate locomotor activity. To evaluate learning, each entrance into the shock zone was recorded as an error. The latency to first enter the shock zone was determined to evaluate between-session place avoidance memory, and the maximum time between entrances to the shock zone was computed to evaluate

within-session place avoidance memory. On day 3, one day after training, the mouse was returned to the arena to assess 24 hr retention of place avoidance memory. The conditions and end-point measures were identical to those used in the training trial on the previous day.

- Conflict Training. The arena was rotating at 1 rpm. On day 3, 1 hr after memory retention was assessed with the shock zone activated, the mouse was returned to the arena with the identical conditions as during initial training, except that the shock zone was relocated 180° to the opposite side of the arena, where the mouse had previously preferred to visit to avoid the shock. There were two 10 min trials to learn the new location of the shock zone with a 1 hr inter-trial interval when the mouse was in its home cage. The same end-point measures were assessed as in the initial learning trials.
- Extinction. The arena was rotating at 1 rpm. On day 4, one day after conflict training, the mouse was returned to the arena under conditions that were identical to the prior training except that the shock was turned off. The mouse received two 10 min trials separated by a 1 hr rest in the home cage, and the same end-point measures were used to evaluate the response to learning that the shock was no longer present.
- Novel environment. The arena was rotating at 1 rpm. On day 6, 2 days after extinction training, the mice were moved to a new apparatus that was located in a different part of the room (different visual environment), and a pad below each grid floor was scented with a drop of vinegar. The mice received a 10 min pretraining session with the shock off and after a 1 hr rest in the home cage they received three 10 min training trials with the shock activated in a 60° sector. The mice rested for 1 hr in the home cage between trials and the same end-point measures evaluated spatial behavior.

## T-maze alternation (L/R discrimination)

The T-maze was constructed with 50 cm tall opaque black-walls on a 40 cm square grid floor. The mice were moved from their cage to the apparatus in a plastic cup and placed in the start arm. They were allowed to explore the T-maze for 1 min and then were removed in the plastic cup. Training began by placing the mouse in the start arm, and, after 5 s, 500 ms, 60 Hz, 0.3 mA shocks began until the mouse escaped to one of the arms that was designated the safe arm. After 10 s in the safe arm, the mouse was removed and returned to the start arm for the next trial. The response was scored as correct if the mouse avoided the shock by escaping within 5 s. The mouse received 10 trials and was then returned to the home cage for a 15 min rest. After the rest, another set of 10 trials began with the safe arm relocated to the opposite side. The mice received a total of four sets of 10 trials with the safe arm alternating left and right between each set of 10 trials. The number of trials required to meet a criterion of two successful escapes was used to evaluate L/R discrimination.

## Acknowledgements

The authors thank Ruth Marx for assistance with hippocampal dissections; Linda Orzolek, Haiping Hao, and David Mohr for assistance with RNA sequencing; Grethe Olsen for assistance with Golgi stains; Yanshu Wang for advice; and James Knierim for advice and helpful comments on the manuscript. Supported by the Howard Hughes Medical Institute (AR, JW, JN), NIH R01NS105472 (CJ, SFH, AF), and a shared instrumentation grant S10 OD023472 (ZH, MA, SM), The Kavli Foundation Centre of Excellence – Centre for Neural Computation, Grant # 227769 of the Research Council of Norway, and the National Infrastructure scheme of the Research Council of Norway – NORBRAIN #197467 (MPW, TK), and NIH grants P50MH094268 and R01MH083728 (CET and MP).

## Additional information

### Funding

| Funder | Grant reference number | Author |
|---|---|---|
| Howard Hughes Medical Institute | | Jeremy Nathans Amir Rattner John Williams |
| National Institute of Neurological Disorders and Stroke | R01NS105472 | André A Fenton Claudia Jou |

| | | Shun Felix Hu |
|---|---|---|
| Shared and High-End Instru-mentation Awards | S10 OD023472 | Susumu Mori |

The funders had no role in study design, data collection and interpretation, or the decision to submit the work for publication.

## Author contributions
Amir Rattner, Chantelle E Terrillion, Menno P Witter, Investigation, Writing - original draft; Claudia Jou, Tina Kleven, Shun Felix Hu, John Williams, Zhipeng Hou, Gloria Shin, Mikhail Pletnikov, Investigation; Manisha Aggarwal, Susumu Mori, Loyal A Goff, Formal analysis; André A Fenton, Formal analysis, Investigation, Writing - original draft; Jeremy Nathans, Conceptualization, Resources, Supervision, Funding acquisition, Writing - original draft

## Author ORCIDs
Amir Rattner (ID) https://orcid.org/0000-0001-9542-6212
Loyal A Goff (ID) https://orcid.org/0000-0003-2875-451X
Menno P Witter (ID) https://orcid.org/0000-0003-0285-1637
André A Fenton (ID) https://orcid.org/0000-0002-5063-1156
Jeremy Nathans (ID) https://orcid.org/0000-0001-8106-5460

## Ethics
Animal experimentation: All mice were housed and handled according to the approved Institutional Animal Care and Use Committee (IACUC) protocol MO16M367 of the Johns Hopkins Medical Institutions.

## Decision letter and Author response
Decision letter https://doi.org/10.7554/eLife.62766.sa1

# Additional files

## Supplementary files
• Supplementary file 1. Sampling parameters chosen for the optical fractionator probe for two hemispheres. Mouse identification numbers are indicated beneath the genotype.

• Supplementary file 2. Area sampling outcomes of the optical fractionator probe for two hemispheres. Mouse identification numbers and left hemisphere (LH) designation are indicated beneath the genotype.

• Supplementary file 3. Stereological analyses of complex spines in CA3. Using an optical fractionator probe, complex spines were classified and counted in the left hemisphere of $Wls^{fl/+}$;$Gfap-Cre$ and $Wls^{fl/-}$;$Gfap-Cre$ mice. The table shows the number of counted spines and, in parentheses, stereologically estimated spines. Five distinct complex spine subtypes were classified: basic, big/prototypical, long, thin, and tall.

• Supplementary file 4. Baseline neurological tests and their outcomes. The data shown here were collected on the cohort of mice tested at New York University (*Figures 6* and *7*). N = 15 $Wls^{fl/+}$; $Gfap-Cre$ mice; N = 14 $Wls^{fl/-}$;$Gfap-Cre$ mice.

• Transparent reporting form

## Data availability
Sequencing data have been deposited in GEO under the accession code GSE157983.

The following dataset was generated:

| Author(s) | Year | Dataset title | Dataset URL | Database and Identifier |
|---|---|---|---|---|
| Rattner A, Terrillion CE, Jou C, Kleven T, Hu SF, Williams J, Hou Z, Aggarwal A, Mori S, Shin S, Goff LA, Witter MP, Pletnikov M, Fenton AA, Nathans J | 2020 | Developmental, cellular, and behavioral phenotypes in a mouse model of congenital hypoplasia of the dentate gyrus | https://www.ncbi.nlm.nih.gov/geo/query/acc.cgi?acc=GSE157983 | NCBI Gene Expression Omnibus, GSE157983 |

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
