## [Decision Letter]

The reviewers acknowledge the scientific rigor and the significant impact of this work. The paper will be of interest to those who are interested in the functional role of the dentate gyrus and the hippocampal circuit.

*Reviewer #1:*

The study by Rattner et al., developed a unique mouse model for studying functional contribution of hippocampal dentate granule cells (DGCs). Using a conditional Wntless (Wls) knockout mouse crossed with Gfap-Cre line, the authors effectively deleted Wls expression in a subset of cortical progenitors. Their anatomical, bulk and sn-RNAseq analysis showed that although most of the cell types in the hippocampus are not affected, neurogenesis of DGCs is largely reduced. Perhaps the most remarkable findings are among the cognitive tests in these mutants - despite an almost complete elimination of DGCs, these mice exhibit only modest behavioral defects. The trace fear conditioning and Barnes maze learning are both normal, while the Morris water maze and active avoidance learning are mildly impaired. Given the significant functional roles of DGCs in pattern separation and cue discrimination proposed by many previous studies, the current study provides a different perspective in functional studies investigating DGCs. Overall, this is a well-executed and highly informative study. I just have a few minor comments.

1) What is the nature of "excitatory neurons" identified in sn-RNAseq? This population of neurons is significantly enlarged in the Wlsfl/-;Gfap-Cre mice. Is it possible that they represent the undifferentiated DGCs? Where are they located in the hippocampus? What is the possibility that this population of neurons is responsible for functional redundancy in the absence of most of DGCs?

2) What happens to EC input to DG in the Wlsfl/-;Gfap-Cre mice? Is it reduced? Does it bypass DGCs and synapse directly onto CA3 pyramidal neurons?

*Reviewer #2:*

This study describes a new mouse model that enables a global analysis of the function of the dentate gyrus. The mouse model - a GFAP-promoter-driven deletion of 'wntless', which causes abolition of wnt signaling in cells with developmental expression of GFAP - produces a 90% reduction of the dentate gyrus. The authors perform a comprehensive analysis of these mice. This study is not only monumental and encyclopedic, but also important and impactful.

The bottom line is: deleting most of the dentate gyrus causes few overall changes in the connectivity of the remaining hippocampal circuits, gene expression in the remaining hippocampal neurons, or any of the studied behaviors. Negative studies are usually frowned upon, but the fact is in my opinion that negative studies are much more important than positive studies. Scores of papers have assigned crucial roles to the dentate gyrus in complex behaviors, especially features of memory. These studies are usually performed with sophisticated methods whose limitations are overlooked by the elegance of their promise. What this paper shows is that these studies are likely misleading, and that the dentate gyrus may, after all, have a more subtle function. I think this is a very important conclusion that is worthy of publication in a premier journal.

The technical quality of the presented data is outstanding. The authors performed excellent and complete analyses that go far beyond what is usually done in systems neuroscience papers. The conclusions are compelling, and I have no doubt after reading this paper that the role of the dentate gyrus is more limited than I previously thought and understood from the many other papers on the subject. Yes, it would have been nice to also do electrophysiology and calcium imaging and optogenetics but this is totally unnecessary - The behavioral, anatomical and RNAseq studies are truly conclusive.

In summary, I recommend publication of this paper without changes or further experimental additions. I think this is an important contribution that is definitive and needs to be out there!

*Reviewer #3:*

This paper provides an extremely thorough analysis of a mouse mutant that lack virtually all of the hippocampal dentate gyrus. Given great interest in the functions of subfields of the hippocampus, this mouse should be helpful for studies of the DG.

The authors have chosen an excellent way to present their mouse mutant to the general community.